# Ozone Response to Emission Reductions in the Southeastern United States

Charles L. Blanchard[1], George M. Hidy[2]

[1] Envair, Albany, CA, 94706, USA

[2] Envair/Aerochem, Placitas, NM, 87043, USA

*Correspondence to*: Charles L. Blanchard (cbenvair@pacbell.net)

**Abstract.** Ozone ($O_3$) formation in the southeastern U.S. is studied in relation to nitrogen oxide ($NO_x$) emissions using long-term (1990s – 2015) surface measurements of the Southeastern Aerosol Research and Characterization (SEARCH) network, U.S. Environmental Protection Agency (EPA) $O_3$ measurements, and EPA Clean Air Status and Trends Network (CASTNet)

nitrate deposition data. Annual 4th-highest daily peak 8-hour $O_3$ mixing ratios at EPA monitoring sites in Georgia, Alabama, and Mississippi exhibit statistically-significant ($p < 0.0001$) linear correlations with annual $NO_x$ emissions in those states between 1996 and 2015. The annual 4th-highest daily peak 8-hour $O_3$ mixing ratios declined toward values of ~45 – 50 ppbv and monthly $O_3$ maxima decreased at rates averaging ~1 – 1.5 ppbv $y^{-1}$. Mean annual total oxidized nitrogen ($NO_y$) mixing ratios at SEARCH sites declined in proportion to $NO_x$ emission reductions. CASTNet data show declining wet and dry nitrate

deposition since the late 1990s, with total (wet plus dry) nitrate deposition fluxes decreasing linearly in proportion to reductions of $NO_x$ emissions by ~60% in Alabama and Georgia. Annual nitrate deposition rates at Georgia and Alabama CastNet sites correspond to 30% of Georgia emission rates and 36% of Alabama emission rates, respectively. The fraction of $NO_x$ emissions lost to deposition has not changed. SEARCH and CASTNet sites exhibit downward trends in mean annual nitric acid ($HNO_3$) concentrations. Observed relationships of $O_3$ to $NO_z$ ($NO_y – NO_x$) support past model predictions of increases in cycling of

NO andincreasing responsiveness of $O_3$ to $NO_x$. The study data provide a long-term record that can be used to examine the accuracy of process relationships embedded in modeling efforts. Quantifying observed $O_3$ trends and relating them to reductions in ambient $NO_y$ species concentrations offers key insights into processes of general relevance to air quality management and provides important information supporting strategies for reducing $O_3$ mixing ratios.

## 1 Introduction

Ozone ($O_3$) is a well-known and important product of photochemical processes in the troposphere involving nitric oxide (NO), nitrogen dioxide ($NO_2$), and volatile organic compounds (VOCs). Ozone is of broad interest for its adverse effects on humans and ecosystems, as reflected by regulation through the U.S. Clean Air Act (e.g., U.S. EPA, 2014; 2015a).

Regulatory actions address extreme $O_3$ mixing ratios: the U.S. National Ambient Air Quality Standard (NAAQS), currently 70 ppbv, is applicable to the annual 4th-highest daily eight-hour maxima averaged over three-year periods (U.S. EPA, 2015b;

2015c). By the early 1990s, U.S. emission control efforts began to focus on nitrogen oxides ($NO_x = NO + NO_2$) in addition to VOCs (NRC, 1991). $O_3$ management has generally relied on precursor reduction requirements estimated from models that integrate descriptions of non-linear chemical and atmospheric processes (e.g., Seigneur and Dennis, 2011), and guidance has also derived from so-called "observation-based" models linking $O_3$ and its precursors based on chemical reactions that are

believed to drive ambient mixing ratios (e.g., NARSTO, 2000; Schere and Hidy, 2000).

Most of the work developing an observational basis for $O_3$-precursor chemistry derives from field campaigns, sometimes focusing on urban conditions. Short-term data are available from aircraft flights, for example, or summer field measurements made at a variety of locations. Such studies usually are limited to a month or two of intense sampling. One example in the southern U.S. is the 1990 ROSE Experiment at Kinterbish, a rural, forested state park in western Alabama (Frost et al., 1998).

This summer study of rural $O_3$ at low anthropogenic VOC and low $NO_x$ mixing ratios provided important insights into rural $O_3$ formation (Trainer et al., 2000). Other examples of short-term campaigns across the U.S. and elsewhere are reviewed in Solomon et al. (2000). More recent field studies include New England in 2002 (e.g., Griffin et al., 2004; Kleinman et al., 2007), Texas in 2006 (e.g., Berkowitz et al., 2005; Neuman et al., 2009), the mid-Atlantic region in 2011 (He et al., 2013), California in 2010 (Ryerson et al., 2013), Colorado in 2012 and 2014 (e.g., McDuffie et al., 2016), and the southeastern U.S. in 2013

(e.g., Neuman et al., 2016; Warneke et al., 2016). These campaigns and accompanying analyses of $O_3$ production and accumulation typically address summer, which historically has the strongest photochemical activity. However, strong photochemical $O_3$ production can occur under special circumstances in winter (e.g., Schnell et al., 2009).

Accounting for an $O_3$ background is important. $O_3$ background is associated with biogenic influence, large-scale transport, or the potential influence of the upper atmosphere (e.g., stratospheric intrusions, especially during spring) (e.g., Lin et al., 2012;

Langford et al., 2015). The nature and magnitude of background $O_3$ remain an active area of research in the U.S. and Europe (Naja et al., 2003; Solberg et al., 2005; Ordóñez et al., 2007; Cristofanelli and Bonasoni, 2009; Arif and Abdullah, 2011; Zhang et al., 2011; Wilson et al., 2012). Hidy and Blanchard (2015) discuss definitions of continental and regional background $O_3$. For this study, we adopt a definition of "background" that includes both the non-anthropogenic component and the southeastern regional component (Section 4.4.1).

Field studies have provided observational evidence of non-linearity in $O_3$-$NO_z$ relationships (e.g., Trainer et al., 1993; Kleinman et al., 1994; Trainer et al., 1995; Hirsch et al., 1996; Frost et al., 1998; Kasibhatla et al., 1998; Nunnermacker et al., 1998; St. John et al., 1998; Sillman et al., 1998; Zaveri et al., 2003; Griffin et al., 2004; Travis et al., 2016). Long-term, post-1990s data are widely available for $O_3$ and $NO_2$ but detailed observations of total oxidized nitrogen ($NO_y$) and VOC, and especially their component species, are typically lacking (e.g., Hidy and Blanchard, 2015). One of the longest records of urban

and suburban data, comprising a series of short-term campaigns as well as continuous measurements, is from southern California. This region exemplifies a photochemically active urban regime. An analysis of multi-decadal (since the 1960s) data by Pollack et al. (2013) reveals how changes in atmospheric chemical reactions have contributed to the observed reductions of $O_3$ in southern California since 1973. Long-term (more than one decade) measurements characterizing $O_3$ and $NO_y$ relationships in both urban and rural conditions are less common.

The photochemical regime in the Southeast represents humid subtropical conditions with urban emissions yielding elevated $O_3$ levels superimposed on a general regional background (Chameides and Cowling, 1995). The EPA $O_3$ and deposition data provide a regional basis for characterizing trends since the early 1980s (U.S. EPA 2016a; 2016b). In addition, the Southeastern Aerosol Research and Characterization (SEARCH) project (Hansen et al., 2003; Hidy et al., 2014) provides measurements that
can be used to investigate changes in $O_3$ production resulting from changes in anthropogenic emissions in the southeastern U.S. The SEARCH network of eight sites began with the Southeastern Oxidant Study (SOS) (Chameides and Cowling, 1995; Meagher et al., 1998) rural locations, which were near (1) Centreville, AL, ~85 km southwest of Birmingham, (2) at Yorkville, GA, ~60 km northwest of Atlanta, GA and (3) at Oak Grove, MS, ~40 km southeast of Hattiesburg, MS, and 75 km north of Gulfport, MS, on private land within the confines of the Desoto National Forest (Hansen et al., 2003). Measurements of some
gas-phase species began at these rural sites in 1992, thus providing a rural data record of over 20 years. Beginning in 1999, SEARCH added five sites in metropolitan Atlanta, GA, Birmingham, AL, Pensacola, FL, and Gulfport, MS.

Our goal for this study is to extend earlier analyses of the photochemical response of $O_3$ to precursors through 2014, emphasizing relationships between $O_3$ and $NO_y$. We first summarize relevant $O_3$ photochemistry to provide a context for the observational analysis. We then describe trends in emissions and ambient pollutant concentrations, and discuss $O_3$, $NO_z$, and
$HNO_3$ observations at the SEARCH sites. The trends in $O_3$ mixing ratio, $NO_x$ precursor emissions, and ambient nitrogen oxide mixing ratios offer important insight into future changes in $O_3$ and $NO_y$. Blanchard et al. (2014) previously explained the majority (66 - 80%) of the day-to-day variations in daily peak 8-hour average $O_3$ at SEARCH sites during March – October of 2002 - 2011 using meteorological variables coupled with ambient measurements of $O_3$ precursors (NO, $NO_2$; limited measurements of VOCs) and $NO_x$ photochemical reaction products ($NO_z$) and a statistical model. The previous analyses are
extended here for data through 2014 to help understand ongoing and potential future $O_3$ changes in relation to changes in ambient $NO_z$ and $HNO_3$ mixing ratios in the southeastern U.S. Results are discussed in relation to modeling predictions by Reynolds et al. (2004) and others.

## 2 Ozone-Nitrogen Oxide Chemistry

### 2.1 Key Atmospheric Reactions Linking $O_3$ with $NO_x$

Net tropospheric $O_3$ accumulation occurs when sunlight acts on VOC and $NO_x$ emissions and the $O_3$ production rate exceeds $O_3$ loss (Trainer et al., 2000). Tropospheric $O_3$ mixing ratios are affected by solar intensity, chemical formation and loss (e.g., deposition) rates of $O_3$, the rate of dispersion of $O_3$ and its precursors, meteorological factors, vertical entrainment and transport of plumes. $NO_2$ forms rapidly by reaction of NO with $O_3$ and photolysis of $NO_2$ produces $O_3$, yielding steady-state mixing ratios of NO, $NO_2$, and $O_3$ in the absence of other species as expressed by the photostationary state, or Leighton relationship
(Seinfeld, 1986).

In the troposphere, $NO_2$ also forms by reaction of NO with peroxy ($HO_2$) and alkyl peroxy ($RO_2$) free radical species, which derive in turn from the reaction of VOCs with hydroxyl (HO), $HO_2$, $RO_2$, and alkyl radicals (Seinfeld, 1986). Radical

production from VOCs creates a pathway for conversion of NO to $NO_2$ that does not consume $O_3$ (Atkinson, 2000), which then leads to higher $O_3$ mixing ratios.

$O_3$ accumulation is typically associated with high solar radiation intensity and temperatures favoring atmospheric reactions, lower wind speeds, and high anthropogenic emission rates (NARSTO, 2000). $O_3$ accumulation requires NO mixing ratios exceeding approximately 10 to 30 pptv (Atkinson, 2000; Logan, 1985), along with the presence of $HO_2$ and $RO_2$ radicals that react with NO to form $NO_2$. The former conditions are normally met in urban air; $NO_x$ mixing ratios are much lower under typical conditions in rural southeastern areas, but still well above 30 pptv (e.g., Hudman et al., 2007; Travis et al., 2016). Under these conditions, the $O_3$ photochemical production rate is proportional to the ambient NO multiplied by the sum of $HO_2$ and $RO_2$ radical mixing ratios, where the latter are weighted by their rates of reaction with NO (Trainer et al., 2000). Field studies show that observed rates of rural $O_3$ production are proportional to the rate of oxidation of $NO_x$. Where VOCs are present for radical production and $NO_x$ is rate-limiting (Trainer et al., 2000), regional $O_3$ production can be expressed in terms of the derivative $d[O_3]/d[NO_x]$, denoted the $O_3$ production efficiency (OPE) (Liu et al.,1987). OPE is understood as the number of molecules of $O_3$ formed per molecule of $NO_x$ oxidized and OPE increases as $NO_x$ mixing ratios decrease (Liu et al., 1987; Trainer et al., 2000). OPE reflects the mean number of $NO$-$NO_2$ cycles occurring, in which each photolysis of one $NO_2$ molecule generates one $O_3$ molecule until that $NO_2$ molecule is oxidized to nitric acid ($HNO_3$) or to other species such as peroxyacetylnitrate (PAN). $NO_x$ reaction products, including $HNO_3$ and PAN, comprise $NO_z$. For chemical reactions, the quantity $d[O_3]/d[NO_z]$ is equivalent to $d[O_3]/d[NO_x]$ but with opposite sign, and has therefore been used to estimate OPE; limitations due to confounding influences of emissions, transport, and deposition are discussed in Sections 4.3 and 4.4.

Empirically, the slope of a linear fit of afternoon $O_3$ (or $O_x = O_3 + NO_2$) versus $NO_z$ has been used to estimate OPE (e.g., Trainer et al., 1993; Pollack et al., 2013). This estimate is subject to certain limitations because it does not explicitly account for: (1) day-to-day variability in "old" (baseline or regional background) $O_3$ mixing ratios, (2) mixing of air masses having different emission histories, (3) rapid loss of $HNO_3$ (primarily through dry deposition, but also through gas-to-particle conversion) (Trainer et al., 2000), and (4) regeneration of $NO_2$ from PAN and certain other species. Because PAN regenerates $NO_2$, it can serve as a reservoir rather than a true $NO_2$ sink (Singh and Hanst, 1981; Singh, 1987). In contrast, $HNO_3$ largely terminates the cycling between NO and $NO_2$. Therefore, the relative yields of PAN and $HNO_3$ are of importance. Despite such limitations in using measurements to quantify OPE, data from field studies have been used since the 1990s to determine upper bounds for OPE and the results have continued to appear in the literature as an indicator of relevance to $O_3$ chemistry (e.g., Berkowitz et al., 2005; Neuman et al., 2009; Kim et al., 2016). Investigators caution that field measurements reveal the net of production and loss, which potentially overestimates actual OPE by factors of 3 to 6 due to rapid chemical and deposition losses of $HNO_3$ and other $NO_z$ species (e.g., Trainer et al., 2000). Additional discussion is found in Section 4.4.

In southern California, changes in the relative proportions of $NO_x$-oxidation products have occurred and are thought to be instrumental in driving the rapid rates of $O_3$ decline in that area (Pollack et al., 2013). These results indicate that measurements of $HNO_3$ or PAN are needed to identify important changes in chemical pathways.

## 2.2 National $O_3$ Response to Emission Reductions

Between 1980 and 2013, the national average of the annual $4^{th}$-highest peak daily 8-hour $O_3$ mixing ratios, a metric relevant to the U.S. $O_3$ NAAQS, declined by 33% (U.S. EPA, 2015d) as national VOC and $NO_x$ emissions decreased by 53% and 52%, respectively (U.S. EPA, 2015e). Across the U.S. and on multiple spatial scales from continental to urban, annual $4^{th}$-highest daily peak 8-hour $O_3$ mixing ratios between 1980 and 2013 show a statistically significant ($p < 0.05$) linear fit to either annual average or to $98^{th}$ percentile daily maximum hourly $NO_2$ mixing ratios; regression slopes are less than 1:1 and intercepts are in the range of 30 to 50 ppbv $O_3$ (Hidy and Blanchard, 2015). Proportionalities between $O_3$ and $NO_2$ that are less than 1:1 are expected, and the observed intercept terms are approximately consistent with typical $O_3$ mixing ratios of ~20 – 50 ppbv observed at remote monitoring sites (Oltmans et al., 2008; 2013; U.S. EPA, 2012; Fiore et al., 2014; Lefohn et al., 2010; 2014; Cooper et al., 2012; 2014).

Although nonlinearity of $O_3$ production and accumulation with respect to ambient VOC and $NO_x$ is well established (Lin et al., 1988), a tendency toward linearity is expected at sufficiently low $NO_x$ mixing ratios. As an example, the $O_3$ photochemical production rate during June 1990 at Kinterbish, AL was approximately linear over a range of ambient $NO_x$ from 0.1 to 2 ppbv (Trainer et al., 2000). Observed $O_3$ extrema can also exhibit an apparent linear or near-linear response to ambient $NO_x$ mixing ratios if the extrema consistently fall within the lower-right quadrant ($NO_x$-sensitive regime) of an $O_3$-VOC-$NO_x$ diagram, a concise graphical representation first established empirically from southern California data and later generated using the Empirical Kinetics Modeling Approach (EKMA) (illustrated in Hidy and Blanchard, 2015). The $O_3$-VOC-$NO_x$ diagram has been adopted by many investigators for displaying the output of box models (e.g., Fujita et al., 2003; 2015) and grid-based photochemical models (e.g., Reynolds et al., 2003; 2004).

Southern California historically has exhibited the highest peak $O_3$ mixing ratios in the U.S. since the 1960s. Because of high ambient $O_3$ and precursor mixing ratios there and the complexity of the relationships of $O_3$ with $NO_x$ and VOC, some investigators have described southern California $O_3$ and precursor trends in terms of percentage changes. For example, Pollack et al. (2013) report that peak 8-hour $O_3$ mixing ratios in southern California declined exponentially over time at a rate of 2.8% per year between 1973 and 2010, thus decreasing $O_3$ levels by approximately a factor of three. This rate of $O_3$ decline exceeds rates occurring in other metropolitan areas (Hidy and Blanchard, 2015). $O_3$ extrema in southern California decreased along with declining mixing ratios of ambient VOCs and $NO_x$ (7.3% $yr^{-1}$ and 2.6% $yr^{-1}$, respectively, 1960 – 2010) and declining ratios of VOC/$NO_x$ (4.8% $yr^{-1}$) (Pollack et al., 2013). The rates of atmospheric oxidation of $NO_x$ increased over time and changes in $NO_x$ oxidation reactions increasingly favored production of $HNO_3$, a $NO_x$ reaction product associated with radical termination and quenching of the $O_3$ formation cycle (Pollack et al. 2013). To our knowledge, changes in the relative proportions of atmospheric reaction products accounting for rapid rates of $O_3$ reduction have not been reported for locations other than southern California.

## 3 Methods

### 3.1 Emissions and Ambient Air Quality Measurements

Air quality monitoring data were obtained from the EPA Air Quality System (AQS) data archives for all sites in Georgia, Alabama, and Mississippi (U.S. EPA, 2016a). Daily measurement values (i.e., peak daily 8-hour $O_3$ mixing ratio) as well as annual summary statistics (e.g., maxima, annual averages) were acquired. We obtained deposition data from the two EPA Clean Air Status and Trends Network (CASTNet) monitoring sites located within the study region: Sand Mountain, AL (125 km ENE of the SEARCH site at Centreville) and Georgia Station, GA (102 km SE of the SEARCH site at Yorkville) (U.S. EPA, 2016b).

Annual, state-level emission trends data were obtained from U.S. EPA (2016c; 2016d), Xing et al. (2013), and Hidy et al. (2014). Comparability of inventories is discussed in the supplementary material (Figure S1). Because the EPA trend inventory utilized different methods for estimating mobile source emissions prior to 2002 compared with 2002 and later years, we combined EPA trend estimates for 2002 – 2016 with the 1996 - 2001 emission estimates of Hidy et al. (2014), which are consistent with more recent EPA methods (supplementary material).

Hourly measurements of gases (NO, $NO_2$, $NO_y$, $HNO_3$, and $O_3$) were obtained from SEARCH public archives (Atmospheric Research and Analysis [ARA], 2017). All parameters measured at the sites are calibrated and audited to conventional reference standards, as described in ARA (2015). Network operations, sampling, and measurement methods are documented in Hansen et al. (2003; 2006); see also Table S1. The network consisted of eight extensively instrumented monitoring sites located in the southeastern U.S. along the Gulf of Mexico and inland (Figure S2): Pensacola, Florida (PNS) and Gulfport, Mississippi (GFP), urban coastal sites (~ 5 km and 1.5 km from the shoreline, respectively); Pensacola – outlying (aircraft) landing field (OLF) and Oak Grove, Mississippi (OAK), non-urban coastal sites near the Gulf (~20 km and 80 km inland, respectively); Atlanta, Georgia – Jefferson Street (JST) and North Birmingham, Alabama (BHM), urban inland sites; and Yorkville, Georgia (YRK) and Centreville, Alabama (CTR), non-urban inland sites. PNS, OAK, and GFP were closed at the end of 2009, 2010, and 2012, respectively. SEARCH site locations are described in detail, including discussion of possible emission influences, in Hansen et al. (2003) and Hidy et al. (2014). SEARCH VOC data are available for JST as daily data from 1999 through 2008, and U.S. EPA VOC measurements are available for YRK as summer hourly data and as 24-hour samples collected every sixth day throughout the year (Blanchard et al., 2010). EPA VOC samples are also available for three other sites in the Atlanta area; only one of these additional sites reported data through 2014.

SEARCH meteorological parameters and gases are sampled at a height of 10 meters, characteristic of lower troposphere mixing ratios near the surface (Hansen et al., 2003; Hansen et al., 2006; Edgerton et al., 2007; Saylor et al., 2010). Gas and meteorological measurements commenced in 1992 at the rural sites of CTR, OAK, and YRK. The measurements at rural SEARCH sites included $O_3$, NO, and $NO_y$ beginning in 1992, and $NO_2$ and $HNO_3$ measurements began in 1996. Consistent measurement methods have been utilized for all gases except $NO_2$. $NO_2$ measurements commenced network-wide in 2002, and three $NO_2$ measurement methods have been employed during the network operations (Table S3). All three methods are

NO$_2$-specific, differing primarily in the light source used for photolysis of NO$_2$. The NO$_2$ data exhibit consistency with NO and NO$_y$ measurements but with some variations occurring during specific years (e.g., 2001 and 2002, Figure S3). Because changes in NO$_2$ measurement methods could affect the computed NO$_z$ (NO$_y$ – NO – NO$_2$), we repeat some data analyses using HNO$_3$ in place of NO$_z$. As noted, HNO$_3$ data also provide useful insight into NO$_2$ termination reactions. HNO$_3$ measurements

are the difference between NO$_y$ and denuded NO$_y$ (Table S1; Hansen et al., 2006). The SEARCH measurements of NO$_y$ were designed to capture particulate nitrate and organic nitrates, as well as NO, NO$_2$, HNO$_3$, and other oxidized nitrogen species. The NO$_y$ sampler derives from the instrument identified in Williams et al. (1998) as "ESE", which was one of five instruments for which measurements of NO$_y$ reproduced the sum of separately measured NO$_y$ species. Additional testing in 2013 showed that SEARCH NO$_y$ measurements agreed with the sum of measured mixing ratios of NO, NO$_2$, HNO$_3$, particulate nitrate, alkyl

nitrates, and peroxy–alkyl nitrates (Hidy et al., 2014).

Trace gas calibrations were done daily for O$_3$ and every third day for other gases. Reported detection limits (Table S1) are 0.05 – 0.1 ppbv for oxidized nitrogen species and 1 ppbv for O$_3$ (Hansen et al., 2003; 2006). NO$_2$ measurement uncertainties are estimated as ~30% prior to 2002 and ~10% after 2002 (Hansen et al., 2006). Measurement uncertainties are estimated to be 10% or less for other oxidized nitrogen species and 5% or less for ozone (2 sigma in all cases). Propagation of errors indicates

corresponding 2-sigma measurement uncertainties averaging 0.5 ppbv for mid-afternoon NO$_z$ (< 0.1 ppbv for NO$_z$ < 1 ppbv) and 0.16 for the ratio NO$_z$/NO$_y$.

## 3.2 Data Analysis

Multiple methods were employed to characterize the variability of ambient O$_3$ and NO$_y$ mixing ratios. Analyses of seasonal variability used data from all months of each year. Diurnal hourly average mixing ratios were computed by year to characterize

patterns of temporal change and to identify hours associated with O$_3$ maxima. Observed slopes of regressions of O$_3$ versus NO$_z$ were computed as previously done in measurement studies using afternoon O$_3$ and NO$_z$ data (Trainer et al., 1993; Kleinman et al., 1994; Trainer et al., 1995; Hirsch et al., 1996; Kasibhatla et al., 1998; Nunnermacker et al., 1998; St. John et al., 1998; Sillman et al., 1998; Zaveri et al., 2003; Griffin et al., 2004; Travis et al., 2016). Because past studies have examined O$_3$ formation in photochemically aged air (i.e., at locations distant from fresh emissions, where atmospheric reactions have

acted on emissions from earlier times) during summers (e.g., Trainer et al., 1993), our analyses focus on the months of June and July to select weeks nearest maximum solar radiation (~ -20 days, + 40 days). Additional analyses were carried out for other months to facilitate comparisons across seasons. As for earlier studies, the calculations are based on afternoon times, using hourly values starting at 2 p.m. local standard time to represent the daily peak O$_3$ after morning production and before mixing ratios decline with decreasing photochemical reaction in later afternoon. In addition to characterizing O$_3$/NO$_z$ and its

change with time, corresponding supporting analyses are presented for O$_3$/HNO$_3$. As a supplemental analysis, rates of maximum diurnal increase of O$_3$ and HNO$_3$ during late morning and early afternoon were computed for comparison of $\Delta$O$_3$ with $\Delta$HNO$_3$.

## 4 Results and Discussion

### 4.1 Trends

Hidy et al. (2014) report a 63% reduction of $NO_x$ emissions in the southeastern U.S. between 1996 and 2014. The largest $NO_x$ emission changes in the Southeast occurred between 2007 and 2009 due to reductions of emissions from electric generating units (EGUs) and from diesel engine vehicles, and were accompanied by more gradual year-to-year reductions of gasoline-engine mobile-source $NO_x$ emissions (de Gouw et al., 2014; Hidy et al., 2014). $NO_x$ emission reductions led to approximately proportional responses of mean ambient $NO_y$ and $NO_z$ mixing ratios at SEARCH sites (Hidy et al., 2014).

The EPA CASTNet data show wet and dry nitrate deposition since the late 1990s declining at rates of ~5% per year (-0.045 ± 0.005 and -0.056 ± 0.005 $y^{-1}$), nearly identical to $NO_x$ emission changes of -0.046 ± 0.001 and -0.051 ± 0.003 $y^{-1}$ (Figure 1). Total (wet plus dry) nitrate deposition fluxes decreased linearly in proportion to reductions of $NO_x$ emissions in Alabama and Georgia (Figure 1). Linear regression slopes indicate that the annual nitrate deposition fluxes at the Georgia and Alabama CASTNet sites correspond to 30% of Georgia emissions and 36% of Alabama emissions on an annual and statewide basis (Figure 1). Emissions are not spatially homogeneous and deposition losses likely vary with distance from emission sources. The two sites are situated differently in relation to metropolitan areas, possibly affecting deposition fluxes; Sand Mountain (SND) is northeast of Birmingham and Georgia Station (GAS) is south of Atlanta. The linearity and statistical significance of the regressions indicates that the fraction of $NO_x$ emissions lost to deposition has not changed over time (ratios of annual deposition-to-state-emissions varied without trend from 0.23 – 0.34 at GAS and 0.30 – 0.45 at SND). Mean annual SEARCH $NO_y$ mixing ratios at rural CTR and YRK declined at ~5 – 7% $y^{-1}$ (Figure S4). SEARCH and EPA CASTNet sites exhibit downward trends in mean annual $HNO_3$ concentrations of ~9 – 11% $y^{-1}$ and ~6 – 7% $y^{-1}$, respectively (Figure S4). Ambient $NO_y$ and $HNO_3$ trends are not statistically different from state-level $NO_x$ emission trends.

Annual 4th-highest daily peak 8-hour $O_3$ mixing ratios at compliance monitoring sites in Georgia, Alabama, and Mississippi exhibit statistically-significant ($p < 0.0001$) linear correlations with annual $NO_x$ emissions in those states between 1996 and 2015 (Figure 2), qualitatively consistent with past work indicating that high $O_3$ would respond to reductions of $NO_x$ emissions (Chameides and Cowling, 1995; Jacob et al., 1995; Kasibhatla et al., 1998). Intersite differences in the annual 4th-highest daily peak 8-hour $O_3$ mixing ratios have decreased (Figure 2), consistent with an analysis of data from a larger number of U.S. and European locations (Paoletti, et al., 2014). The annual 4th-highest daily peak 8-hour $O_3$ mixing ratios are declining toward non-zero values, as indicated by the statistically-significant ($p < 0.0001$) intercepts of ~45 – 50 ppbv (Figure 2). SEARCH data are used to characterize the southeastern $O_3$ response to emission changes in greater detail. Between 1999 and 2014, the highest peak daily 8-hour $O_3$ mixing ratios occurring each month (monthly $O_3$ maxima) declined at all SEARCH sites at statistically significant ($p < 0.01$) rates averaging ~1 – 1.5 ppbv $y^{-1}$ (Figure 3). These declines are comparable to the trend in the 95th percentile summer peak daily 8-hour $O_3$ mixing ratios in the southeastern U.S. of ~ -0.8 to -1.8 ppbv $yr^{-1}$ reported by Lin et al. (2017), with downward trends occurring in other seasons as well. The observed SEARCH $O_3$ trends are also consistent with other analyses of North American observations (e.g., Chan, 2009; Lefohn et al., 2010; Paoletti, 2014; Simon et al., 2015) and

with the trends occurring at EPA monitors in the Southeast (Figure 2). Both EPA (Figure 2) and SEARCH (Figure 3) data suggest that $O_3$ mixing ratios increased during the 1990s, then began declining. This result is consistent with modeling by Reynolds et al. (2004), which predicted an initially slow response of Atlanta-area $O_3$ to emission reductions between 1996 and 2000, followed by increasing sensitivity to reductions of $NO_x$ emissions. The observed $O_3$ decreases exceed those predicted

by Reynolds et al. (2004) because actual $NO_x$ emission reductions (~60% between 1996 and 2014) exceeded the projected emission reductions (~40% by 2010 and ~55% by 2020) that were anticipated based on known and anticipated emission control rules at the time of the modeling study. The SEARCH trends are compared with emission changes in the Southeast, and with emission and $O_3$ trends in southern California, in Table S2.

More complete understanding of regional $O_3$ trends requires consideration of both regional emission changes and possible

changes in background $O_3$. Multiple definitions of the term "background $O_3$" may be found in the literature, including global background, continental background, non-anthropogenic background, and regional background, among others. For the $O_3$ trends shown in Figures 2 and 3, the most relevant consideration is the amount of $O_3$ transported into the study domain across upwind boundaries (denoted here as regional background or transported $O_3$). The percentage reductions of $O_3$ are larger if transported $O_3$ can be estimated and subtracted from observed $O_3$ mixing ratios, and this adjustment potentially provides a

better assessment of the effects of regional emission reductions on the fraction of $O_3$ that is manageable by means of local and regional emission control measures. For example, Parrish et al. (2017a) report that the $O_3$ enhancement above background in Southern California decreased by 4.5% $yr^{-1}$, which is larger than the unadjusted $O_3$ decline of 2.8% $y^{-1}$ given by Pollack et al. (2013).  Similarly, rates of decline in southeastern U.S. $O_3$ are larger if regional background $O_3$ is considered (Table S2).

Defining and estimating regional background (or transported) $O_3$ are each challenging. We interpret the intercepts in Figure 2

as indicators of mean $O_3$ levels that would occur on days with weather conducive to high $O_3$ in the absence of $NO_x$ emissions from AL and GA sources, i.e., as estimators of $O_3$ transported into the region from outside the study domain (as discussed subsequently, multi-day carryover of local and regional emissions during stagnation events could also affect intercepts and slopes). Days with weather that is not conducive to high $O_3$ likely have different levels of transported $O_3$. The statistically-significant slopes in Figure 2 indicate $O_3$ enhancements that are attributable to AL-GA emissions, except as noted next, and a

comparison of the $O_3$ decline to intercept-corrected $O_3$ would then reveal the proportionality between AL-GA emissions and AL-GA $O_3$ enhancements over $O_3$ originating outside the study domain (i.e., in excess of regional background $O_3$). Although the ~30 – 35% $O_3$ declines are less than proportional to the ~60% decrease in $NO_x$ emissions, the decline in the median $O_3$ is ~60% if the 50 ppbv intercept shown in Figure 2 is subtracted from the $O_3$ mixing ratios.

If the amount of $O_3$ that has been transported from upwind regions has been changing over time, e.g., declining as $NO_x$

emissions and ambient $O_3$ decline in adjacent regions, the slopes shown in Figure 2 would reflect changes in both the $O_3$ that originated upwind and in the $O_3$ enhancements attributable to AL-GA emissions, confounding attribution. Related studies do not provide consistent evidence for a trend, either upward or downward, in regional background $O_3$ in the southeastern U.S. For example, baseline $O_3$ concentrations in air flowing into Texas from the Gulf of Mexico during May through October did not change significantly between 1998 and 2012 (Berlin et al., 2013). Mean regional background $O_3$ mixing ratios were 48

ppbv to 59 ppbv in the Houston, TX, area on days with $O_3$ levels exceeding the NAAQS, which includes $O_3$ contributions from transport to the area from other regions of the U.S. (Berlin et al., 2013). Observed trends in the 5th percentile $O_3$ have previously been used as indicators of changes in either regional or continental background $O_3$ (e.g., Wilson et al., 2012). The 5th percentile peak daily 8-hour $O_3$ mixing ratios decreased during summer at rural sites throughout the southeastern U.S. between 1988 and

2014 (Lin et al., 2017). By this measure, regional background $O_3$ levels were not increasing in the southeastern U.S. during our study period.

Large-scale transport affecting $O_3$ in the boundary layer and at the surface is a function of altitude. For example, during June 2013, anthropogenic emissions and long-range transport (long-range tropospheric + stratospheric) $O_3$ each accounted for about 40% (15 – 20 ppbv each) of model-predicted $O_3$ below 1 km altitude at Huntsville, AL, while long-range transport accounted

for ~80% of model-predicted $O_3$ above 4 km altitude (Johnson et al., 2016). This variation of source contributions with altitude provides an opportunity to differentiate between emission-related and transport-related trends derived from vertical soundings of upper-air $O_3$ mixing ratios. Using ozonesondes that are generally launched on a weekly schedule, vertical $O_3$ mixing ratio profiles have been determined by the University of Alabama in Huntsville, Alabama, since 1999 (Newchurch et al., 2003; Johnson et al., 2016; University of Alabama, 2017; NOAA, 2017). We obtained these ozonesonde data (n = 940 days) and

identified the following statistically significant trends in the lower layers that are relatively more influenced by local and regional emissions according to Johnson et al. (2016): $-0.25 \pm 0.11$ ppbv $y^{-1}$ (p < 0.05) in daily measurements at 0.5 km, $-0.40 \pm 0.10$ ppbv $y^{-1}$ (p < 0.0001) at 1 km (daily), $-0.42 \pm 0.09$ ppbv $y^{-1}$ (p < 0.0001) at 2 km (daily), and $-0.57 \pm 0.13$ ppbv $y^{-1}$ in monthly averages of $O_3$ measurements made throughout the interval 1 – 2 km  (p < 0.001). At higher altitudes where Johnson et al. (2016) predicted that long-range transport is the dominant source of $O_3$, no trends occurred: $0.06 \pm 0.08$ ppbv $y^{-1}$ (p >

0.1) at 4 km (daily) and $0.09 \pm 0.19$ ppbv $y^{-1}$ (p > 0.1) at 8 km (daily).

Global background is one component of regional background and trends in global background are expected to contribute to trends in regional background. Lin et al. (2017) show that rising $NO_x$ emissions in Asia have increased modeled North American background $O_3$ levels (based on model simulations with zero North American emissions) by ~0.2 ppbv $yr^{-1}$ in the southeastern U.S. in summer, which is a small effect even when cumulated over 20 years in comparison with the ~25 ppbv

reduction in the median annual 4[th]-highest peak daily 8-hour $O_3$ shown in Figure 2. Multiple studies have demonstrated increasing trends in global background $O_3$ mixing ratios (Ordóñez et al., 2007; Oltmans et al., 2008; Arif and Abdullah, 2011; Wilson et al., 2012). Parrish et al. (2017a) report that the highest $O_3$ design values (the 3-year running mean of the annual 4[th]-highest peak daily 8-hour $O_3$ mixing ratio) in Southern California are converging toward of limit of $62.0 \pm 1.9$ ppb, which they identify as the $O_3$ design values that would result from U.S. background $O_3$ concentrations. Parrish et al. (2017b) report

decreasing $O_3$ transported across the Pacific into the western U.S. after 2000. As noted, regional background $O_3$ in the southeastern U.S. does not appear to be trending either upward or downward, even though trends in background $O_3$ have been established in other areas or globally.

In the southeastern U.S., the simple conceptual model of $O_3$ transported into a study region across upwind boundaries is incomplete. High $O_3$ typically occurs during multi-day stagnation episodes, which are associated with the presence of high

barometric pressure over the domain and limited transport (Blanchard et al., 2013). Transport distances determined from 24-hour back-trajectory computations are less then 300 km for the highest decile $O_3$ (Blanchard et al., 2013). Mean 24-hour transport distances are less than 350 km during June and less than 380 km during July (Blanchard et al., 2014). These distances are approximately equivalent to distances from Birmingham to Mobile, AL, or from Atlanta to Savannah, GA. Local and

regional emissions can accumulate over multiple days and potentially could contribute to observed $O_3$ concentrations (e.g., aloft) that are considered as regional background. In contrast to emissions originating upwind, carryover from emission sources within the study domain is a manageable component of efforts to reduce $O_3$.

## 4.2 Seasonal Variations of $O_3$, $NO_y$, $NO_z$, $HNO_3$, and VOCs

The seasonal oscillations of monthly $O_3$ maxima in the Southeast are coupled to local or regional meteorology, solar radiation, and emissions (e.g., Blanchard et al., 2013; 2014; Hidy et al., 2014). Variations of daily maximum temperature and mid-day relative humidity (RH)are associated with variations of daily peak 8-hour $O_3$ mixing ratios by ~ ±30 percent from mean peak 8-hour $O_3$ mixing ratios, after also accounting for variations of other meteorological factors (Blanchard et al., 2014). Air mass back trajectories originating from the south (~ 150 to 200 degrees) exhibit peak 8-hour $O_3$ that is ~5 – 10 percent lower than

average; daily peak $O_3$ decreases as 24-hour back trajectory distances increase from zero to ~600 km, consistent with association of higher $O_3$ concentrations with air mass stagnation rather than transport (Blanchard et al., 2013; 2014). At SEARCH sites, the monthly $O_3$ maxima (highest daily peak 8-hour $O_3$ each month) and mean daily peak 8-hour $O_3$ mixing ratios typically occurred in summer months, especially inland, and declined more than other monthly maxima (Figures 3 and 4). Summer means were not always higher than spring averages, especially at rural and coastal sites and during more recent

years (Figure 4). Roughly constant winter monthly peak 8-hour maxima of ~40 ppbv occurred throughout the period of record (Figure 3). The seasonal variability of the highest peak daily 8-hour $O_3$ therefore declined over time (see also Table S3). Similar results were found for monthly means of hourly measurements, discussed in Section 4.3 on diurnal variations. Other recent studies have reported decreasing seasonal variability of $O_3$ across the U.S. using data from large numbers of monitoring sites (Chan, 2009; Chan and Vet, 2010; Cooper et al., 2012; Paoletti et al., 2014; Simon et al., 2015). Declines in seasonal variability

are thought to result from changing rates of $O_3$ formation as precursor emissions have declined, or from increasing influence of intercontinental background $O_3$, not from changes in seasonal variations of temperature and other meteorological factors (Chan, 2009; Cooper et al., 2012; Simon et al., 2015).

The SEARCH data indicate that seasonal variations occur in ambient $O_3$, $NO_y$, $NO_z$, $HNO_3$, and the ratio of $NO_z/NO_y$ (Figure 5). Seasonal variations of temperature and other meteorological factors are known to cause seasonal variations of $O_3$ and $NO_y$

species concentrations. The monthly average $NO_z$ and $HNO_3$ mixing ratios indicate that active photochemical processing of $NO_x$ occurs during well more than half the year in the warm climate of the southeastern U.S.The effects of VOC species on $O_3$ formation depend on both their ambient concentrations and their reactivities. To describe VOC variations at sites with long-term VOC measurements, we use isoprene data as an indicator of biogenic VOCs and toluene as an indicator of anthropogenic

VOCs (nominally emitted as a gasoline vapor). The importance of isoprene emissions for $O_3$ production in the southeastern U.S. is well established (e.g., Chameides et al., 1988; Chameides and Cowling, 1995; Frost et al., 1998; Starn et al., 1998; Wiedinmyer et al., 2006; Zhang et al., 2014; Lin et al., 2017). We also consider other reactive VOC species of interest, including α-pinene (biogenic) as well as ethylene and xylenes (anthropogenic). Summer (June – August) months exhibit

elevated ambient mixing ratios of rural and urban isoprene, typically about 5 – 10 ppbC, that are one to two orders of magnitude greater than those occurring between October and April (Figure 6). Transitions between low and high ambient isoprene mixing ratios occur in mid-May and mid-September in northern Georgia (Figure 6). Annual mean isoprene mixing ratios were relatively constant, ~2.5 – 3 ppbC, between 1998 and 2014. Biogenic VOCs, primarily isoprene, represent ~20% of the VOC reactivity at JST, ~30% at South Dekalb (SDK, located in metropolitan Atlanta ~16 km southeast of JST), and ~50% at YRK,

averaged over all samples collected between 1999 and 2007 (Blanchard et al., 2010a). Through precursor interactions, seasonal variations in isoprene mixing ratios are expected to affect seasonal variations in $O_3$ mixing ratios and production rates.

Mean mixing ratios of ethylene and aromatic compounds vary substantially between urban and rural sites and exhibit less, and a different, seasonal variation than does isoprene, peaking in the fall rather than in the summer (compare Figures 6, S5, S6). Daily average mixing ratios of toluene, xylenes, and ethylene decline over the years, consistent with regulatory reductions of

anthropogenic VOC emissions (Figures S5, S6). Seasonal variations in ambient mixing ratios and trends in the anthropogenic emissions of aromatic compounds are expected to influence $O_3$ mixing ratios and production in urban settings (rural anthropogenic VOC mixing ratios are lower but detectable).

The 24-hour average VOC mixing ratios are of somewhat limited value for showing the influence of VOCs on $O_3$ formation and accumulation. VOC influence is dependent on $NO_x$ mixing ratios, which vary depending on proximity to emission sources

and time of day. Meteorological variability, including diurnal and day-to-day changes in temperature, vertical mixing, cloud cover, photolysis, and air mass transport, further obscures the quantitative effects of VOCs on seasonal and interannual variations of $O_3$. Influences of anthropogenic VOCs at SEARCH sites have previously been reported (Blanchard et al., 2010b; 2014) and are not analyzed beyond this summary.

## 4.3 Diurnal Variations of $O_3$, $NO_y$, $NO_z$, and $HNO_3$

Summer (June – August) mean $O_3$ mixing ratios exhibit characteristic nocturnal minima and mid-day (noon to 4 p.m., midpoint ~ 2 p.m.) maxima at all SEARCH sites (Figure 7). This diurnal pattern remained essentially the same at both the urban and rural sites from 1999 through 2014, but the daytime maxima decreased. Between 1999 and 2014, the summer mean mid-day maxima declined by ~30 ppbv at all sites, while nocturnal means exhibited variable responses (Figure 7). Similar diurnal variations occur throughout the year, with smaller decreases in the mean mid-day $O_3$ maxima occurring during seasons other

than summer (Figures S7 – S9). By the end of the study period, diurnal $O_3$ profiles were higher during spring (March through May) than summer at the rural sites (CTR and YRK, Figures S7 and S8), consistent with the reduction in summer mean monthly daily peak 8-hour $O_3$ averages (Figure 4). Decreasing summer diurnal mean $NO_y$, $HNO_3$, and $NO_z$ mixing ratios were also observed, with a general flattening of the profiles and with the times of maxima remaining consistent (Figures S10-12).

$O_3$ changes are discussed in relation to changes in $NO_y$ and $NO_z$ in Section 4.4, with emphasis on summer and additional consideration of spring months.

## 4.4 Observed Relationships between $O_3$ and $NO_z$

As discussed above, $O_3$ mixing ratios vary seasonally and diurnally in response to variations in emissions, weather, background
$O_3$, and other factors. To reduce the influence of seasonal and diurnal variability, this section focuses on mixing ratios of $NO_z$, $HNO_3$, and $O_3$ at 2 p.m. during June and July. Both temperature and solar radiation are typically high during June and July, and multi-day stagnation events occur frequently in association with high barometric pressure (Blanchard et al., 2013). Exceptions exist during the passage of frontal systems (Blanchard et al., 2013; Figure S13). The 2 p.m. hour has the highest, or close to highest, average hourly $O_3$ for all sites and years (Figure 7). The atmosphere is well-mixed by mid-day. Over the
range of ambient mixing ratios observed across 15 years, the June-July 2 p.m. $O_3$ values are distinctly nonlinear in relation to ambient $NO_z$ and $HNO_3$ mixing ratios (Figure 8). More variability is evident at urban sites than at rural sites, consistent with influence of urban $NO_x$ and perhaps VOC emissions on $O_3$. The nonlinearity indicated in Figure 8 is also evident when the data are restricted to days having the highest peak daily 8-hour $O_3$ mixing ratios (Figure S14).

### 4.4.1 Linear Models

Linear regressions are fit to the afternoon data by year, as shown in Figure 9 for 2013 and in Table S4 for all years. During multi-week periods within any summer, all sites exhibit near-linear relationships of mid-day $O_3$ to $NO_z$. Because the ranges of $NO_x$ and $NO_z$ mixing ratios within each year are limited, year-specific relationships are close to linear and linear models are statistically significant. Steeper slopes at rural sites than at urban sites in Figure 9 suggest that either more $O_3$ molecules formed per molecule of $NO_x$ consumed in rural locales than in urban areas, or that greater losses of $NO_z$ occurred at the rural sites, as
discussed below. At all sites, similar results are obtained for regressions of $O_x$ ($O_3 + NO_2$) vs $NO_z$ compared with $O_3$ vs $NO_z$ (Figure 9, caption). At 2 p.m., rural $O_3$ mixing ratios are nearly identical with $O_x$ mixing ratios and with other metrics (e.g., $O_3$ – [$NO_y$ – NO]) (Figure S15). At urban sites, 2 p.m. $NO_2$ mixing ratios are non-negligible, but this difference alters the intercepts rather than the slopes of the regressions of $O_x$ vs $NO_z$ compared with $O_3$ vs $NO_z$ (Figure 9). As previously noted (Figure S13), even during the two-month periods that we analyzed, the weather is not always conducive to $O_3$ formation and such days could
influence the observed slopes and intercepts. However, regression results restricted to days with weather that favors $O_3$ formation (as defined in Figure 9) do not differ from the unrestricted regressions.

Plotting the year-specific (June – July) computed regression slopes versus mean June – July 2 p.m. $NO_z$ shows significant increases over time as ambient $NO_z$ mixing ratios have decreased, subject to year-to-year variability (Figure 10, Table S4). Similar urban-rural differences and patterns of increasing regression slopes are also observed when data are restricted to March
and April (spring) at YRK and JST (Figure S16). The results for spring show more variability than the summer year-specific linear models. One key difference between spring and summer days is that cumulative solar radiation between sunrise and 2

p.m. is greater on summer days than on spring days, presumably fostering greater photochemical extent of reaction and accumulation of $O_3$ during summer.

The regression slopes determined from 2 p.m. data could reflect day-to-day differences in transported $O_3$ if background $O_3$ is consistently higher on high-$O_3$ days than on low-$O_3$ days and $NO_z$ is not (in contrast, random variations in day-to-day background $O_3$ and $NO_z$ would introduce variations, or scatter, around the regression lines). We checked for an effect of this type by repeating the analyses using differences in mixing ratios. Two sets of difference-based regressions are used: (1) the differences between 2 p.m. and 10 a.m. hourly measurements, and (2) the differences between 11 a.m. and 10 a.m. hourly measurements. The differences are computed for each day to minimize or eliminate the unknown day-specific background levels, and are then used in the regressions. These hours were selected to focus on times of day when the atmosphere is well-mixed. The morning rise in mixing heights is expected to contribute to increases in the mixing ratios of secondary species as aged air aloft is incorporated into the mixed layer. The most rapid rates of increase in diurnally-averaged $O_3$, $NO_z$, and $HNO_3$ values occur between ~8 a.m. and 12 noon local time (Figures 7, S8 – S9). By mid- to late-morning hours during summer, considerable vertical entrainment has occurred, and subsequent changes in the mixing ratios of secondary species likely reflect same-day atmospheric chemical reactions. Computing afternoon – morning differences and late morning – mid-morning differences helps account for day-to-day variations in regional background $O_3$, but also introduces higher relative uncertainties because four measurements (two differences) are used in the regressions. Results for all three approaches are tabulated in Table S5, by site and year. Like the regressions based on 2 p.m. measurements, the difference-based regressions indicate that observed slopes have increased over time (Table S5). The difference-based regressions exhibit lower slopes than the non-differenced afternoon regressions, which could be due to lesser statistical fit, or to better accounting for regional background $O_3$, or to a combination of these factors. The difference-based regressions suggest that $O_3$-$NO_z$ slopes increased from less than 5:1 in the late 1990s and early 2000s to values between 5:1 and 10:1 after 2010 (Table S5). These lower slope values are consistent with our previous results in which observed$O_3$-$NO_z$ relationships were determined while also accounting for day-to-day variations in meteorology, which indicated that JST, YRK, and CTR $O_3$/$NO_z$ slopes were 3.5, 5.0, and 7.1, respectively, within the range of 1 to 5 ppbv $NO_z$, for measurements made during March – October of 2002 - 2011 (Blanchard et al., 2014).

A second potential effect on the temporal changes in the regression slopes could be due to changes in $NO_2$ measurement methods, previously described; this possibility was checked by using regressions of $O_3$ vs. $HNO_3$ (Figure S17). The results indicate that the relationship in Figure 10 is not an artifact of changes in $NO_2$ measurement methods. The record is more complete for the regressions of $O_3$ vs. $HNO_3$, because the $HNO_3$ measurements were made over a longer time than the $NO_2$ measurements (and the latter are needed for computing $NO_z$). As shown for YRK, the year-specific slopes of 2 p.m. $O_3$ vs. $NO_z$ and for $O_3$ vs. $HNO_3$ each increased substantially after about 2008 (Figures 10, S17). The $O_3$ vs. $NO_z$ and $O_3$ vs. $HNO_3$ regression slopes tend to level out after 2011, and possibly decrease somewhat, but variability is too high to project beyond the observed data ranges (Figures 10, S17). Similar results are obtained for spring for JST and YRK (Figures S18 and S19).

Our increases in year-specific slopes of $O_3$ versus $NO_z$ potentially could be due to increasing losses of $NO_z$ species, especially $HNO_3$, over the long-term SEARCH record. As previously noted, however, the CASTNet data show declining rates of both

wet and dry nitrate deposition since the late 1990s, with no change in the ratio of deposition to emissions (Figure 1). Therefore, the long-term slope increases cannot be attributed to increasing deposition losses of $HNO_3$ (whether absolute or fractional). Qualitatively, the CASTNet data suggest that the observed slopes would likely be at least a factor of two smaller if adjusted for deposition losses. This adjustment would be comparable to the 1990s studies discussed in Section 4.4.2.In Figure 9, the intercepts of year-specific regressions for 2013 approach 20 ppbv $O_3$, which could be interpreted as a regional background $O_3$ level relatively unaffected by local chemistry. These values are lower than those in Figure 2 and lower than the estimated range of 48 ppbv to 59 ppbv for air transported into the Houston area. They are also lower than modeled western non-U.S.-anthropogenic regional background $O_3$ levels of ~ 40 – 50 ppbv (Lefohn et al., 2014; Dolwick et al., 2015) but are consistent with model estimates of non-U.S.-anthropogenic background $O_3$ less than ~30 ppbv in Atlanta (Lefohn et al., 2014). Since regression intercepts restricted to days with weather that favors $O_3$ formation do not differ much from the intercepts of the unrestricted regressions (Figure 9), our low intercepts for recent years do not appear to be linked to meteorological conditions that specifically favor $O_3$ loss over formation. However, when considered over the full set of years, the $O_3$-$NO_z$ relationships on the highest $O_3$ days differ from those on larger subsets of the data (Figure S14). Possibly, the intercept terms cannot be fully interpreted without additional consideration of $O_3$ carryover in multiday episodes, as previously noted. The intercept terms for earlier years are higher than for later years; for example, the intercepts for the YRK regressions range from $27 \pm 3$ to $42 \pm 4$ ppbv prior to 2009 (for all but two of these years, intercepts are 36 – 38 ppbv). The intercept terms for earlier years are consistent with 1997 – 2006 eastern U.S. summer baseline $O_3$ levels ($32 \pm 12$ ppbv in the absence of continental influences) reported by Chan and Vet (2010).

Higher intercepts during early years could be due to fitting a linear regression to the upper portion or the mid-range of the nonlinear relationship between $O_3$ and $NO_z$, as shown in Figures 8 and S14. The nonlinearity and the downward trends in mean $NO_z$ and $HNO_3$ mixing ratios mean that slopes of regressions computed at higher mean $NO_z$ and $HNO_3$ mixing ratios should not be extrapolated beyond their range of applicability to the y-intercept. Alternatively, the trend toward lower intercepts could reflect declining mixing ratios upwind of the study sites, consistent with documented long-term reductions of ambient $O_3$ mixing ratios throughout the U.S. (e.g., Chan and Vet, 2010; Lefohn et al., 2010; Paoletti, 2014; Simon et al., 2015; Hidy and Blanchard, 2015). As previously discussed, however, regional background $O_3$ in the southeastern U.S. does not appear to be trending either upward or downward,

### 4.4.2 Comparisons with Observational and Modeling Studies

The preceding section demonstrates that the slopes of the regressions of $O_3$ versus $NO_z$ increased over time and examines the potential influence of measurement artifacts, weather, deposition, and pollutant transport on the results; none of these plausible influences adequately explains why the slopes of the regressions of $O_3$ versus $NO_z$ increased. The increasing slopes appear to indicate that relationships between $O_3$ and $NO_z$ changed over time, yet the physical processes associated with the changes remain ambiguous. Modeling studies offer insights. Modeling process analysis by Reynolds et al. (2004) for the eastern U.S. predicted that the number of NO cycles (i.e., the ratio of new plus recreated NO to new NO) would increase from 8 to 14

(~75%) in central (metropolitan) Atlanta and from 9 to 11 (~20%) northwest of Atlanta in response to a 60% reduction of $NO_x$ emissions from a 1996 emissions base case. Both the modeled emission reduction (60%; compare to Figure 2) and the modeling subregions (central Atlanta, JST; northwest Atlanta, YRK) are directly comparable to our study period and domain. NO cycling is relevant to our regressions of $O_3$ versus $NO_z$ because an $O_3$ molecule is produced, with some loss, each time NO cycles

through a set of reactions until NO cycling terminates in reactions products that are components of $NO_z$ (e.g., $HNO_3$, PAN). Thus, the observed increases in the slopes of the regressions of $O_3$ versus $NO_z$ are directionally consistent with modeling predictions, but are larger than the predicted 20 – 75% increases in NO cycling.

The data were selected to represent periods that have consistent weather from day to day to minimize the influence of meteorological variability, and regressions of subsets of the data yield slopes and intercepts comparable to those based on all

days of June and July (Figure 9). However, the observed $O_3$ decreases that have occurred in the region (Figures 2 and 3) could not have occurred if $O_3$ formation rates increased by factors of ~3 to 4 (as suggested by Figure 10), or even by a factor of two (Table S5), since $NO_x$ emissions declined by ~60%, or ~5% per year over 20 years (Figure 1; Hidy et al., 2014). This consideration suggests that increased NO cycling, while likely linked with our observational results, cannot be the only factor involved. The regression slopes are nonetheless consistent with related studies when a basis for comparison exists.

The SEARCH observed afternoon slope values of ~5:1 prior to 2003 – 2007 are comparable to, or lower than, similar regression results obtained in studies during the 1990s, which showed observed summer slope values of 11:1 in rural Georgia in 1991 (Kleinman et al., 1994), 8.5:1 at rural eastern sites (Trainer et al., 1993), 7:1 near Birmingham, AL in 1992 (Trainer et al., 1995), 5.7:1 near Nashville, TN in 1995 (Sillman et al., 1998), and 4.7:1 near Nashville, TN, in 1999 (Zaveri et al., 2003), and to modeling results and observations with composite regression slope values of 6.7 and 7.6, respectively, within the afternoon

planetary boundary layer in the eastern U.S. during the summer of 2002 (Godowitch et al., 2011) The SEARCH regression slope values prior to 2003 – 2007 are, as expected, higher than other 1990s values that were corrected for deposition losses, which, for example, yielded adjusted estimated values between 3:1 and 5:1 near Nashville in 1995 (Nunnermacker et al., 1998; St John et al., 1998; Sillman et al., 1998). Our higher observed slope values after 2010 are consistent with aircraft measurements made in the Southeast in August and September 2013, which show $O_x$ (= $O_3$ + $NO_2$) versus $NO_z$ slope of 17.4, and they are

also consistent with model calculations, which show slopes of 14.1 to 16.7 (Travis et al., 2016). Consistent with our regressions, Travis et al. (2016) did not adjust for variations in background $O_3$ and $NO_z$. For comparability, we note that our $O_3$ versus $NO_z$ regression slopes were 13.1 to 18.8 (± 1.2 to 1.4) in June and July, 2013, at three of four sites (25.7 ± 2.8 at the fourth site, which is the most rural in character) and our $O_x$ versus $NO_z$ slopes were 12.0 to 18.9 (± 1.2 to 1.4) at three of the four sites (25.8 ± 2.8 at the fourth site). The increase in recently observed slope values that we report is therefore supported by the 2013

data of Travis et al. (2016). Our apparently high regression slope values are comparable to observations that averaged 12.9 in ship plumes and 33.5 in assumed background marine air, as reported by Kim et al. (2016) using data from a 2002 study of ship emission plumes off the coast of southern California, though the specific conditions associated with these two studies are different from ours and thus limit the applicability of the comparisons.

The increase in regression slopes with decreasing ambient $NO_x$ and $NO_z$ is also directionally consistent with computations by Liu et al. (1987), which showed increasing OPE as $NO_x$ declines. The numerical results of the modeling calculations by Liu et al. (1987) are specific to the modeled conditions, which represented complete oxidation of VOCs over a period of months. However, increases in model-predicted $NO_x$ OPE with declining $NO_x$ results from multiple factors, such as radical reactions involving VOCs and $NO_x$, that are pertinent to other situations (Lin et al., 1988).

In contrast to southern California, where Pollack et al. (2013) reported a shift from PAN to $HNO_3$ production, the SEARCH data do not definitively show a changing fraction of $HNO_3$ relative to $NO_y$. Increasing formation of PAN (which regenerates $NO_2$) and decreasing formation of $HNO_3$ (which terminates cycling between NO and $NO_2$) could facilitate $O_3$ accumulation as ambient $NO_x$ and $NO_z$ mixing ratios continue to decline. Since the long-term SEARCH data record does not include measurements of PAN, this possible effect could not be investigated.

## 5 Implications

The trends in, and relationships between, $O_3$ and $NO_y$ species provide some insight into the potential for future $O_3$ changes in the southeastern U.S. The post-1990s $O_3$ trend provides one guide to future average rates of $O_3$ reduction in the sense that the rates of $O_3$ reduction during the next decade are unlikely to deviate dramatically from those of the recent past. Anthropogenic $NO_x$ and VOC emissions are each expected to continue to decline. Anthropogenic VOC mixing ratios have declined since 1999, but natural components such as isoprene and terpene mixing ratios have remained relatively constant (Figure 6; Blanchard et al., 2010a; Hidy et al., 2014), leaving ambient VOC levels increasingly dependent on biogenic emissions. Evidence suggests that $O_3$ formation in the SEARCH region will move toward more $NO_x$ sensitive conditions with continued decreases in $NO_x$ and anthropogenic VOC emissions, coupled with high levels of natural VOC emissions in the region. This anticipated emission reduction path should reinforce the $O_3$-$NO_z$ relationships and the interpretation presented here.

## 6 Conclusions

The seasonal differences in the past $O_3$ response to $NO_x$ emission reductions could have implications for future $O_3$ management if spring and autumn $O_3$ maxima fail to decline and thereby become a focus of concern that merits attention comparable to summer $O_3$ maxima. The observed relationships of $O_3$ to $NO_z$, which is the product of reactions involving $NO_x$, are nonlinear and suggest increasing responsiveness of $O_3$ to $NO_x$ over the study period. In addition, changes in the relative importance of chemical reactions that yield $HNO_3$ compared with PAN are likely to play a role in altering $O_3$ accumulation. Long-term documentation and analysis of trends in $O_3$ mixing ratios in relation to $NO_x$ emission reductions and decreases in ambient reactive nitrogen ($NO_y$) concentrations yields opportunities for obtaining insights about ambient $O_3$ reductions that complement and corroborate air quality modeling predictions and add substantially to the "weight-of-evidence" approach for air quality management adopted by the U.S. government after 2000.

**Data Availability**

The SEARCH data are available at https://www.dropbox.com/sh/o9hxoa4wlo97zpe/AACbm6LetQowrpUgX4vUxnoDa?dl=0. EPA data are available at http://aqsdr1.epa.gov/aqsweb/aqstmp/airdata/download_files.html and at https://www.epa.gov/castnet.

**Author Contributions**

C. L. B. and G. M. H. designed the study and wrote the manuscript. C. L. B. carried out the statistical analyses.

**Competing Interests**

The authors declare that they have no conflict of interest.

**Acknowledgments**

The authors thank Atmospheric Research and Analysis, Inc. for collecting, validating, and providing SEARCH data, and J. Jansen for managing the SEARCH study. Funding for the SEARCH network was provided by Southern Company with contributions from the Electric Power Research Institute. Southern Company provided partial financial support for analysis of SEARCH data. We are indebted to these sponsors for supporting this unique long-term measurement program.

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

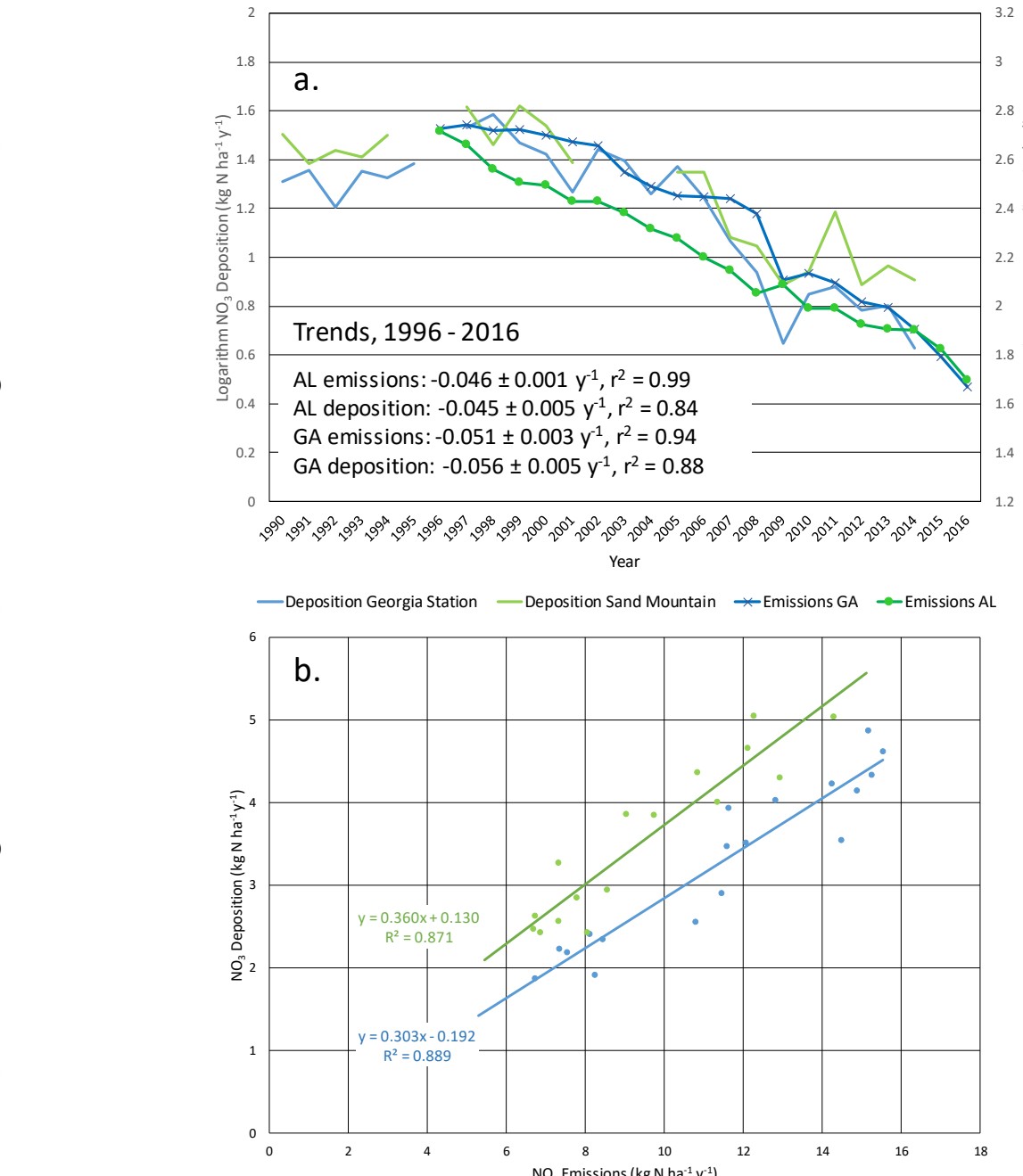

**Figure 1. Comparison of nitrate deposition (wet plus dry) to NO$_x$ emission densities in Georgia and Alabama as (a) temporal trends and (b) regression of deposition against emissions (with same color coding in both panels). Nitrate deposition and NO$_x$ emission densities are expressed as kg ha$^{-1}$ y$^{-1}$. NO$_x$ emissions are from all source sectors (supplement). Panel (a) shows natural logarithms vs. year and indicates that emissions and deposition trended downward at the same rates. Panel (b) slopes are statistically significant ($p < 0.0001$) and intercepts are not ($p > 0.1$).**

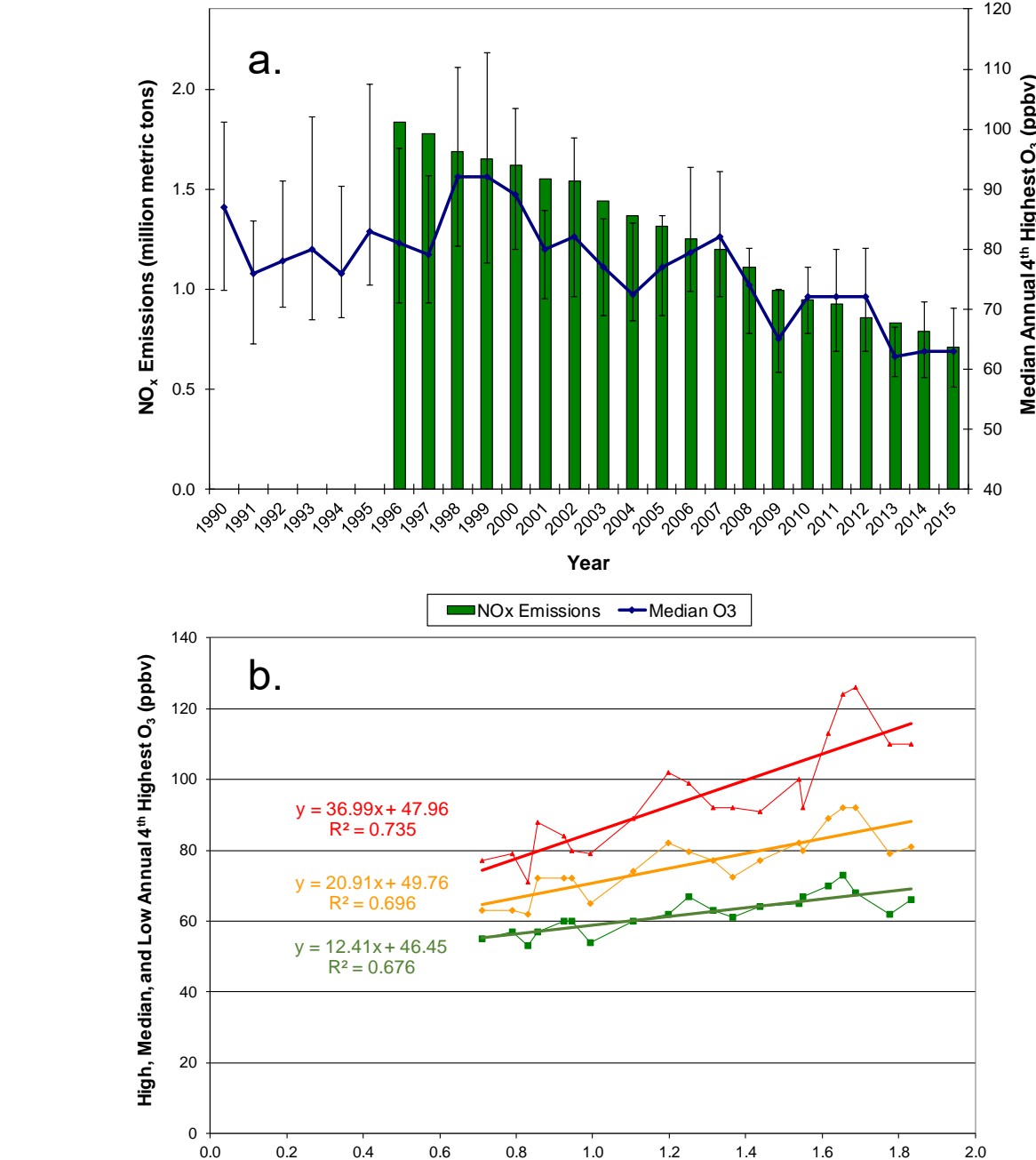

**Figure 2. Comparison of annual 4th-highest daily peak 8-hour $O_3$ to $NO_x$ emissions in Georgia and Alabama (a) trends (± 90th and 10th percentile sites) and (b) regressions (high = 90th percentile site, median, and low = 10th percentile site annual 4th-highest daily peak 8-hour $O_3$). $NO_x$ emissions are from all source sectors (supplement). $O_3$ data include all EPA AQS monitors in Georgia and Alabama for each year having at least 75% data completeness (mean = 55 monitors, low of 32 – 36 in 1990 – 1993). Slopes and intercepts are statistically significant (p < 0.0001).**

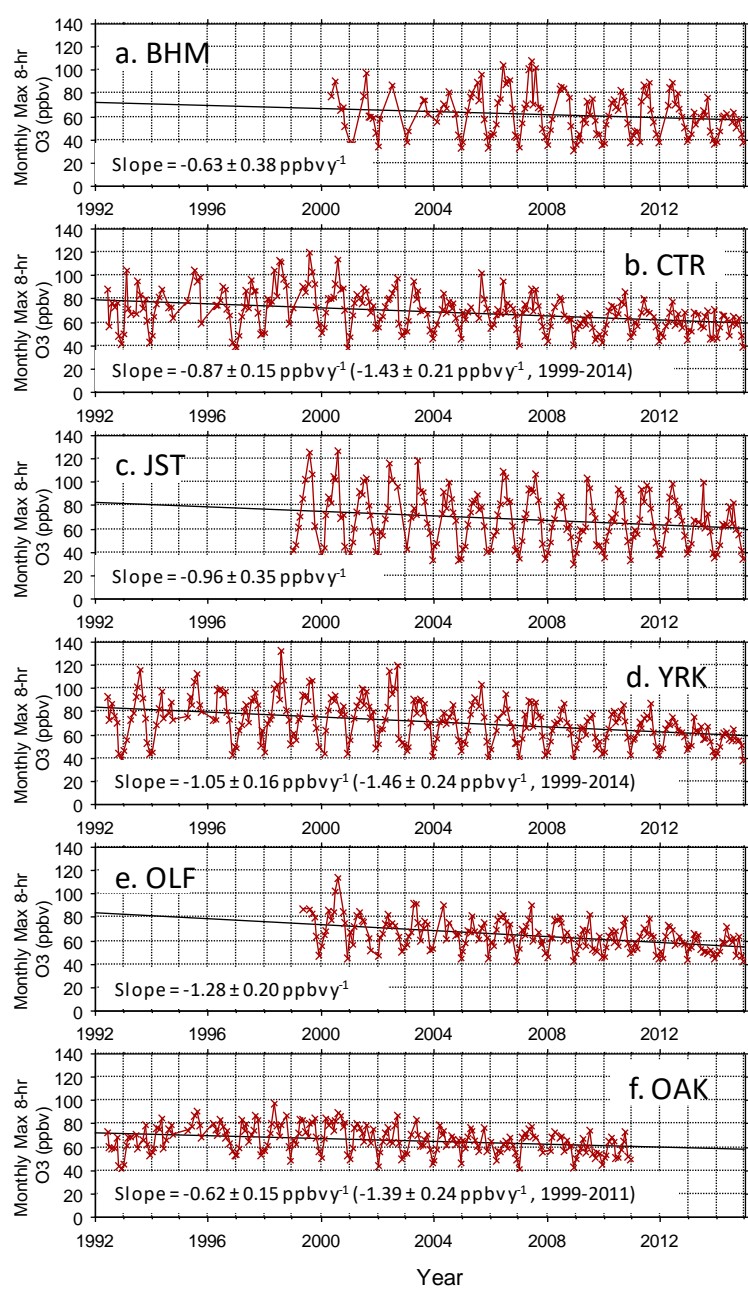

**Figure 3. Monthly maxima of daily peak 8-hour average $O_3$ mixing ratios. All monthly maxima are determined from 24 or more days with 18 or more sampling hours per day. PNS and GFP (not shown) exhibit trends of -1.64 ± 0.45 and -0.60 ± 0.32 ppbv y$^{-1}$, respectively. Trends are statistically significant (p < 0.01) at CTR, JST, OAK, OLF, PNS, and YRK.**

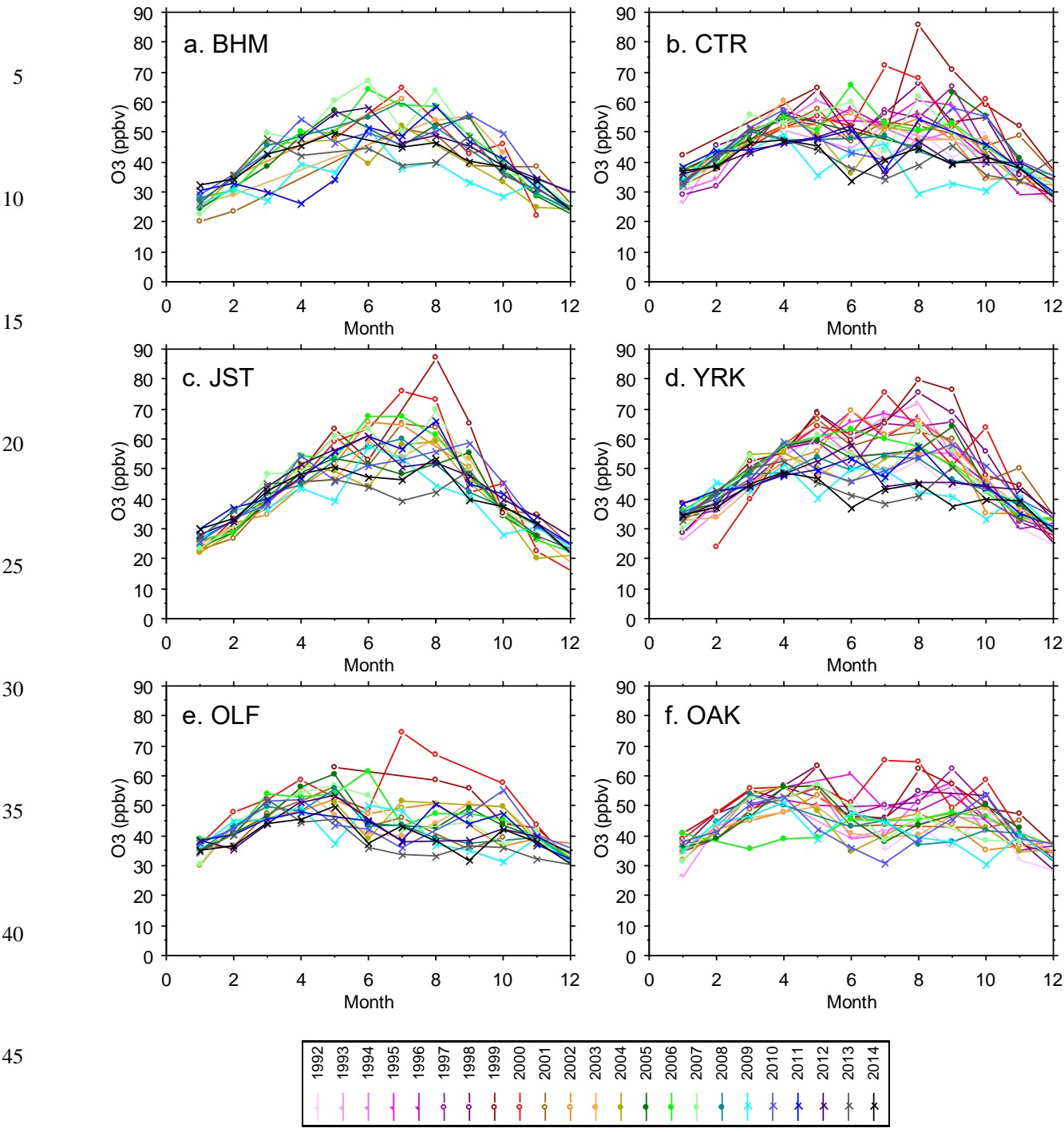

**Figure 4. Monthly means of daily peak 8-hour average O$_3$ mixing ratios. All monthly means are determined from 24 or more days with 18 or more sampling hours per day. Standard errors of the means average 2 (range 0.8 – 5) ppbv.**

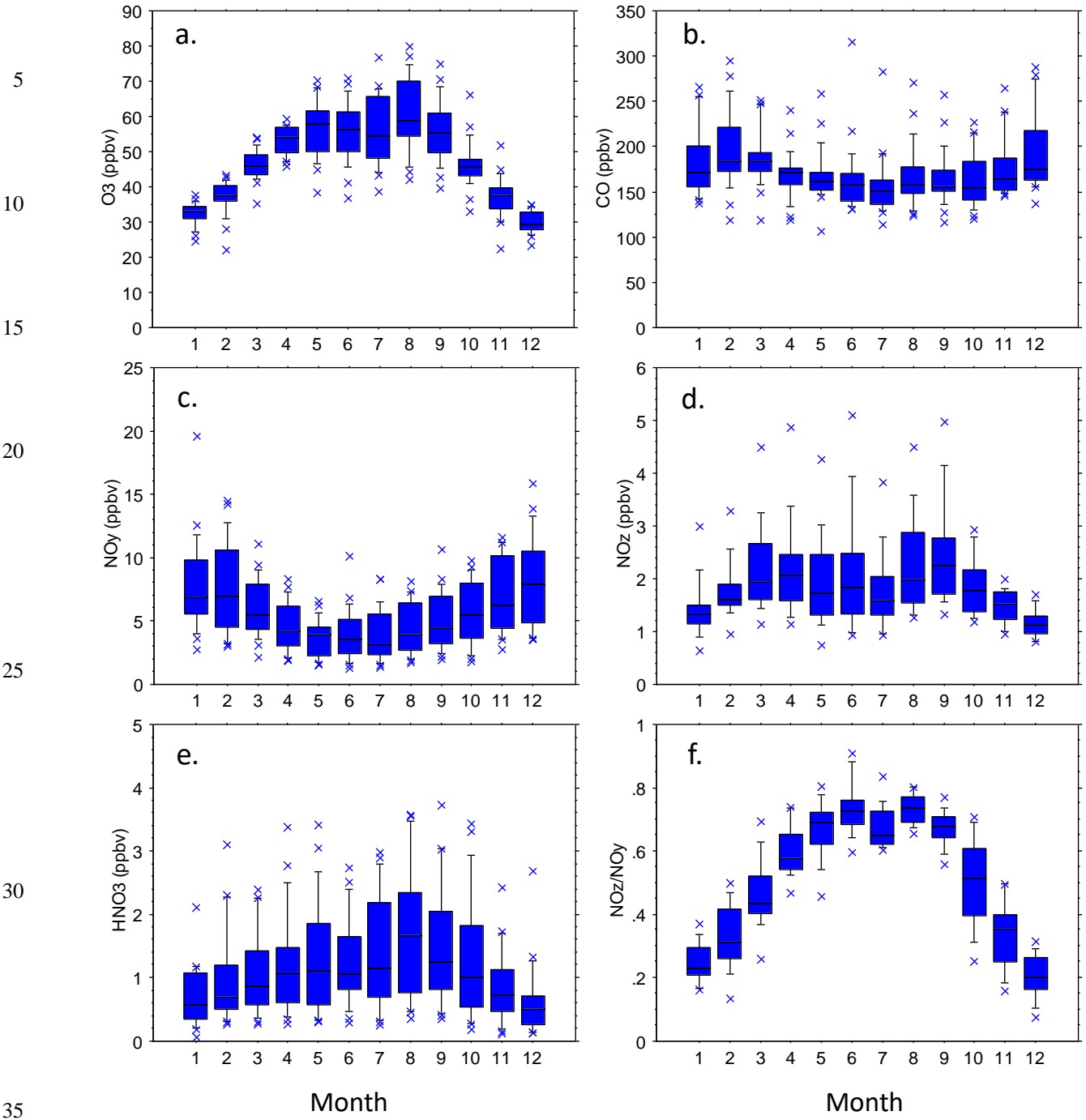

**Figure 5. Statistical distributions of mean monthly species mixing ratios, all SEARCH sites, 1992 – 2014. Distributions indicate the 10th, 25th, 50th, 75th, and 90th percentiles of the monthly averages.**

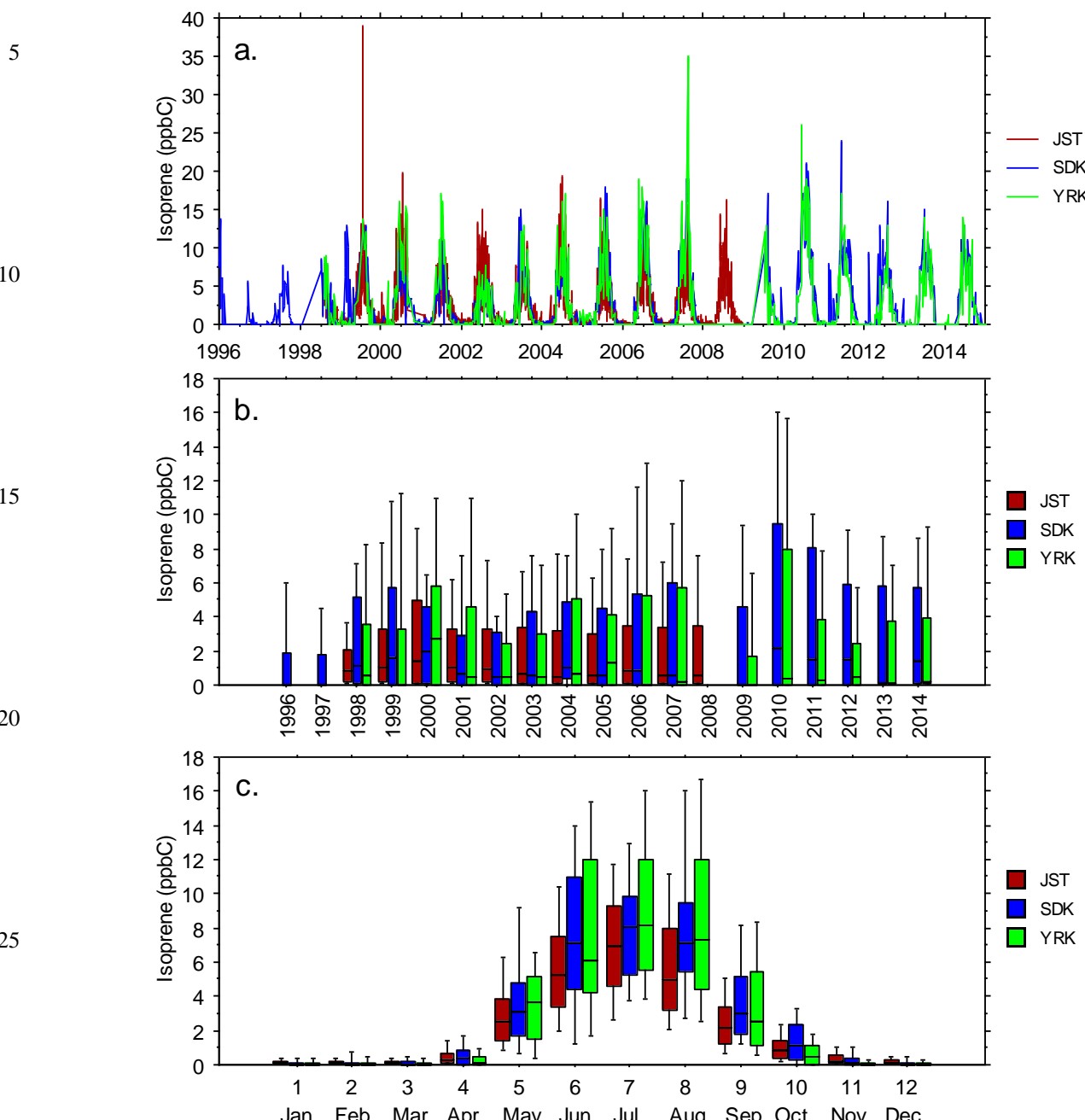

**Figure 6. (a) Daily-average isoprene mixing ratios vs. date, (b) statistical distributions of daily-average isoprene mixing ratios vs. year, and (c) statistical distributions of daily-average isoprene mixing ratios vs. month. Samples were obtained every day at JST and once every six days at YRK and SDK (Blanchard et al., 2010). Distributions indicate the 10th, 25th, 50th, 75th, and 90th percentiles.**

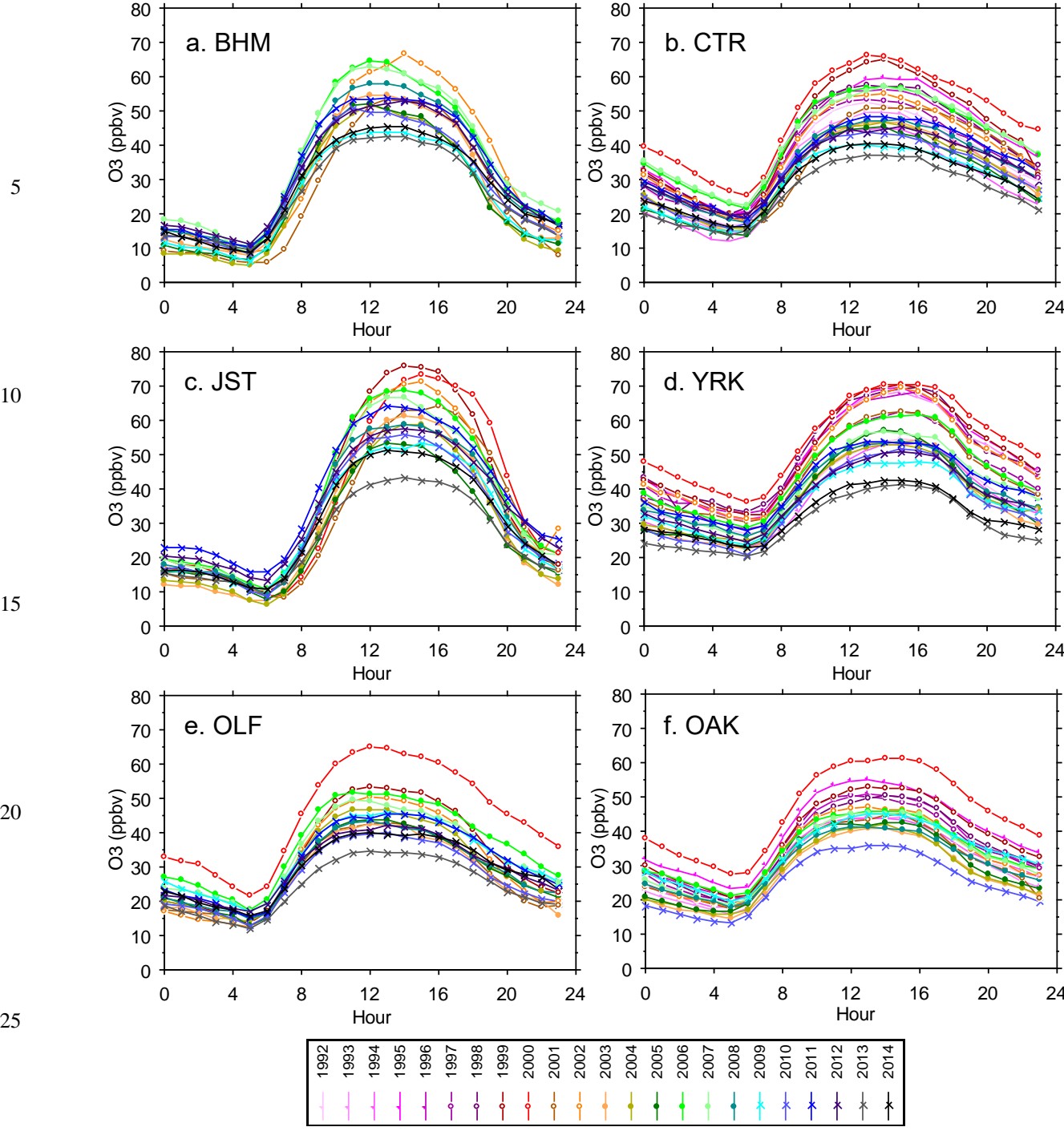

**Figure 7. Average O₃ mixing ratios vs. hour, by year. Each data point is the mean of all hourly measurements during June through August. Sites at PNS and GFP (not shown) exhibit similar diurnal profiles and trends (sampling at those sites ended after 2009 and 2012, respectively). Standard errors of the means are 0.3 – 4 ppbv, ~2% of mean O₃ mixing ratios.**

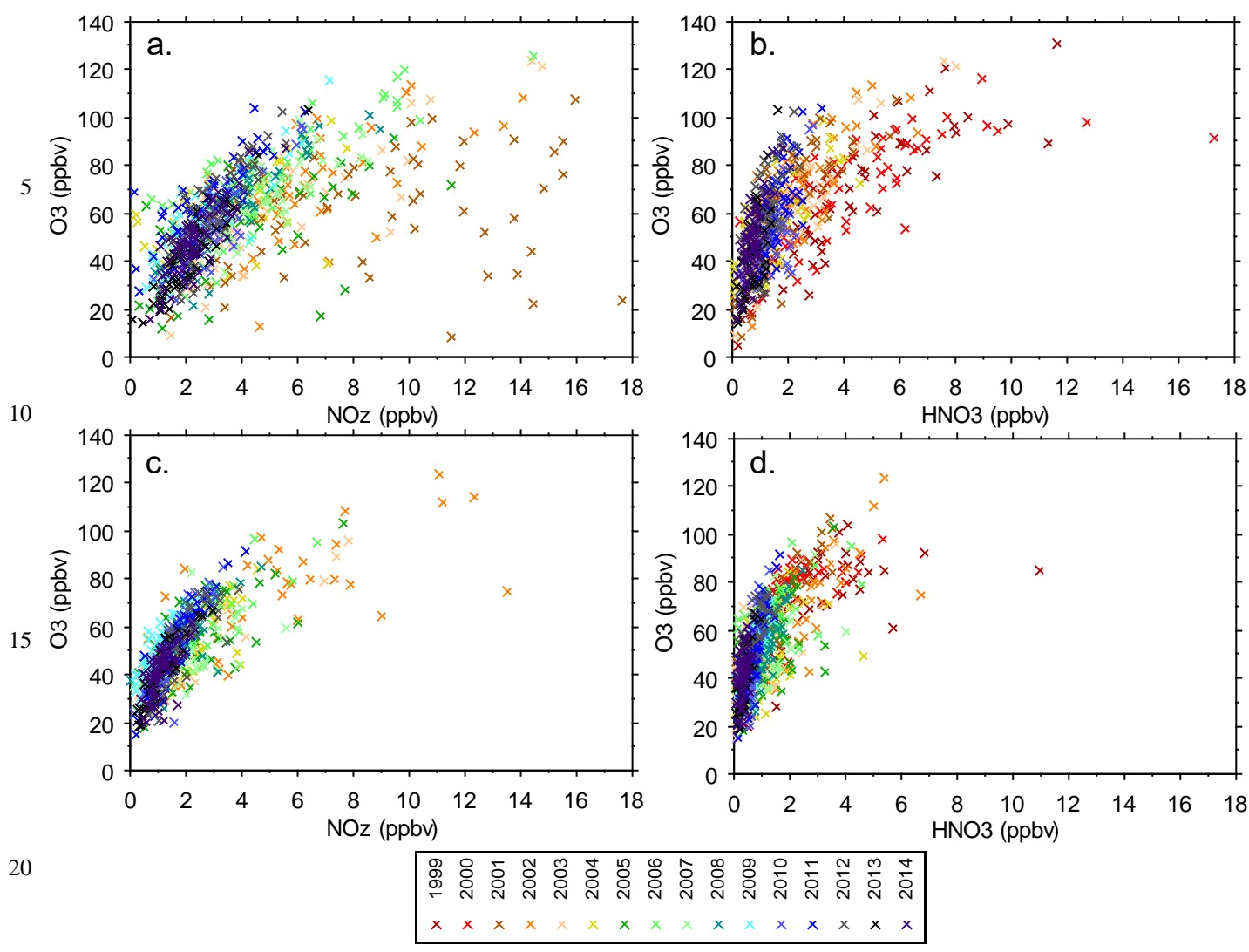

**Figure 8.** (a) $O_3$ vs. $NO_z$ at JST; (b) $O_3$ vs. $HNO_3$ at JST; (c) $O_3$ vs. $NO_z$ at YRK; and (d) $O_3$ vs. $HNO_3$ at YRK. Each point is the 2 – 3 p.m. hourly average on one day, limited to days in June or July and delineated by year. The 2001 and 2002 $NO_z$ data may be biased high due to lower $NO_2$ mixing ratios obtained by the instrumentation used at that time (Figure S2).

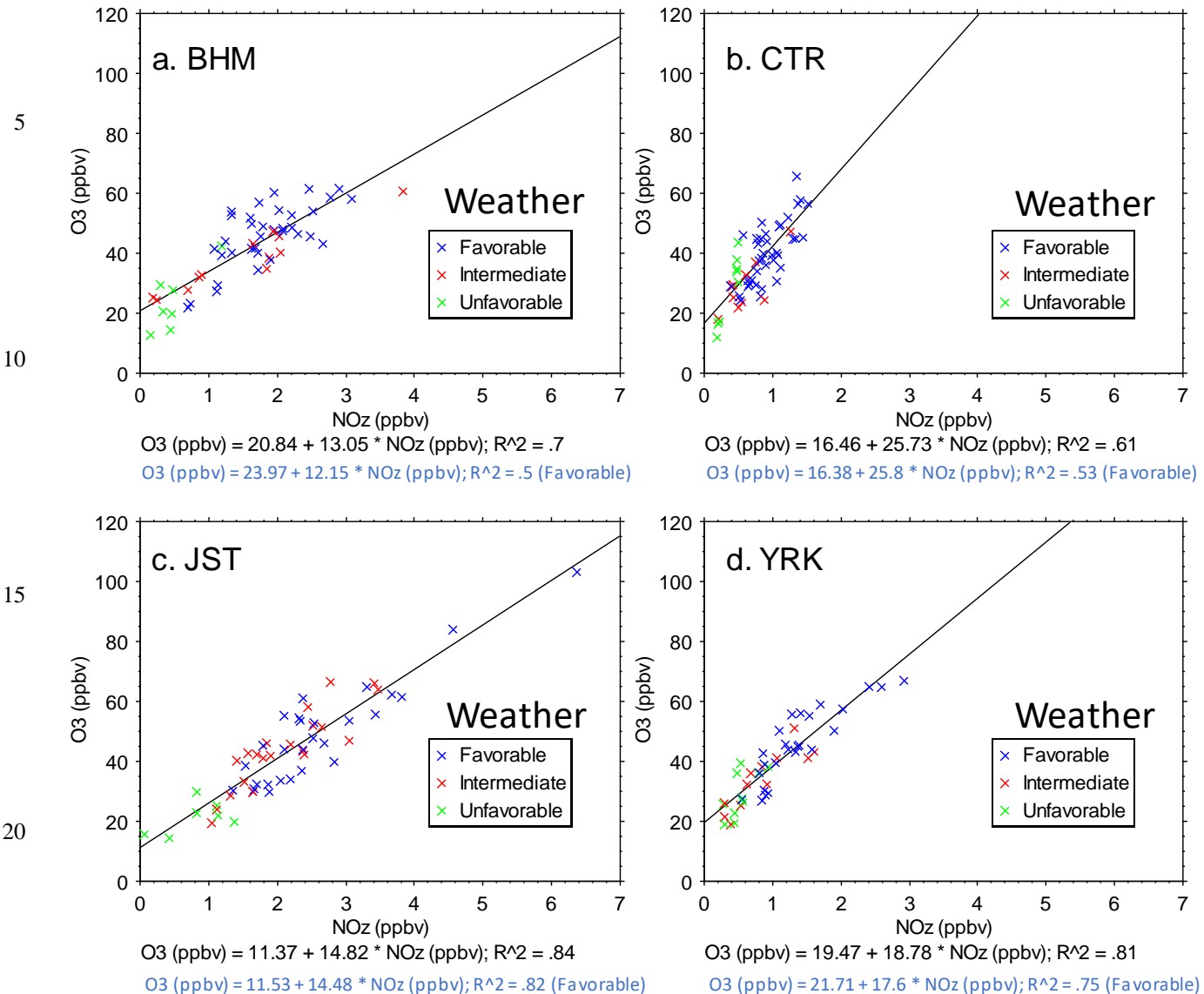

O3 (ppbv) = 20.84 + 13.05 * NOz (ppbv); R^2 = .7
O3 (ppbv) = 23.97 + 12.15 * NOz (ppbv); R^2 = .5 (Favorable)

O3 (ppbv) = 16.46 + 25.73 * NOz (ppbv); R^2 = .61
O3 (ppbv) = 16.38 + 25.8 * NOz (ppbv); R^2 = .53 (Favorable)

O3 (ppbv) = 11.37 + 14.82 * NOz (ppbv); R^2 = .84
O3 (ppbv) = 11.53 + 14.48 * NOz (ppbv); R^2 = .82 (Favorable)

O3 (ppbv) = 19.47 + 18.78 * NOz (ppbv); R^2 = .81
O3 (ppbv) = 21.71 + 17.6 * NOz (ppbv); R^2 = .75 (Favorable)

**Figure 9. O$_3$ vs. NO$_z$ during June and July, 2013. Each point is the 2 – 3 p.m. hourly average on one day. The data were selected to represent the approximate mid-point of the mid-day O$_3$ maxima and to span a period around the summer solstice (- ~20 days, + ~40 days) when solar radiation is highest on average. The regression slopes show higher rural than urban values: BHM = 13.05 ± 1.19 ppbv ppbv$^{-1}$, JST = 14.82 ± 0.88 ppbv ppbv$^{-1}$, YRK = 18.78 ± 1.38 ppbv ppbv$^{-1}$, CTR = 25.73 ± 2.76 ppbv ppbv$^{-1}$. Corresponding regression slopes for O$_x$ vs. NO$_z$ are: BHM = 12.00 ± 1.16 ppbv ppbv$^{-1}$, JST = 13.88 ± 0.93 ppbv ppbv$^{-1}$, YRK = 18.85 ± 1.37 ppbv ppbv$^{-1}$, CTR = 25.79 ± 2.79 ppbv ppbv$^{-1}$. Symbols indicate the favorability of weather to O$_3$ formation and accumulation: (1) favorable = T > 25°C, RH < 70%, and solar radiation > 500 W m$^{-2}$, (2) intermediate = neither favorable nor unfavorable, (3) unfavorable =  T < 25°C, RH > 70%, and solar radiation < 500 W m$^{-2}$. Regression results are shown for all days and for the days with favorable weather.**

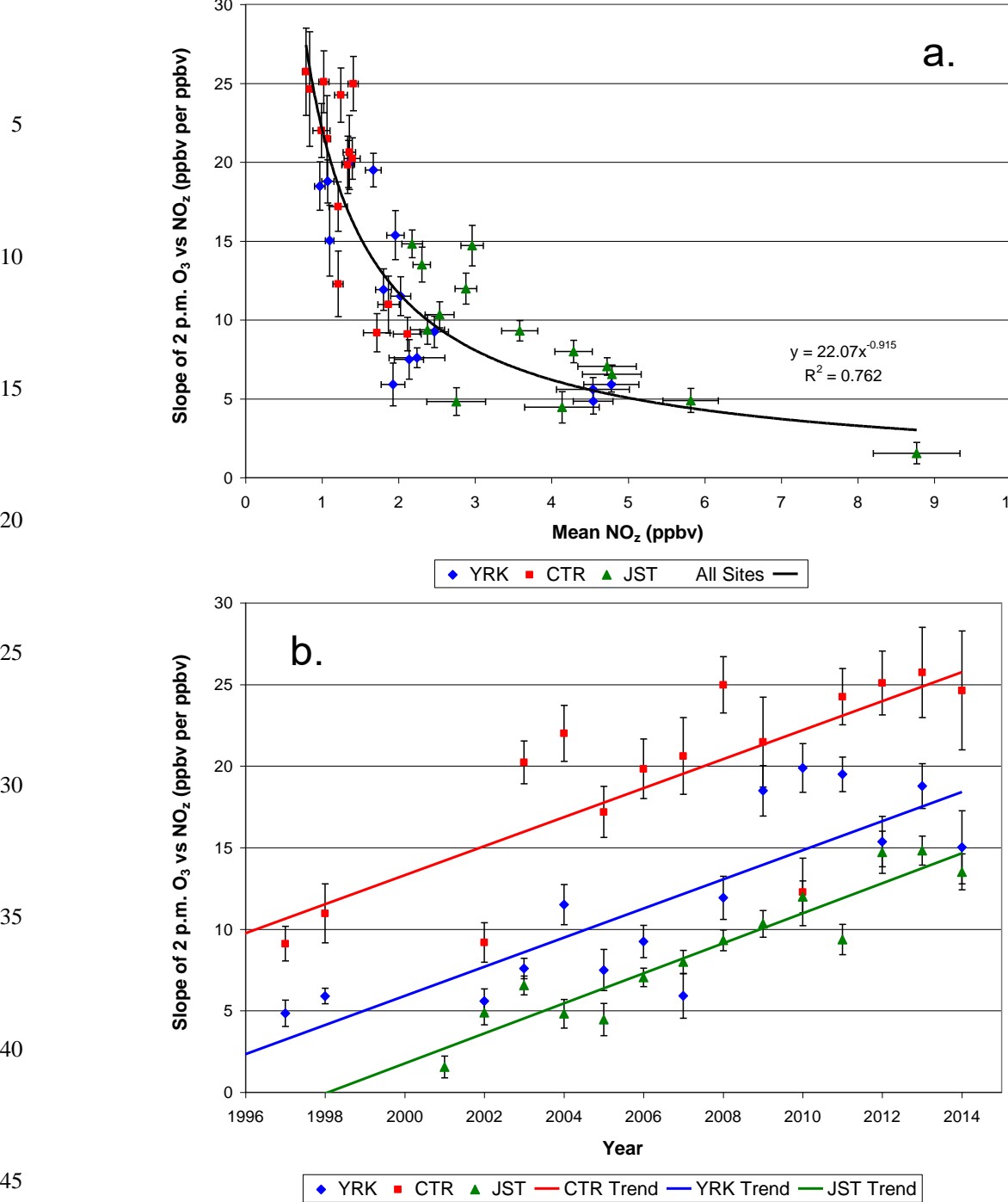

**Figure 10. (a)** Summer CTR, JST, and YRK slope of daily (2 p.m.) $O_3$ and $NO_z$ vs. mean (2 p.m.) $NO_z$ mixing ratios, and **(b)** summer regression slope vs. year. $NO_2$ data were not available for 1999 through 2001. Vertical and horizontal error bars are one standard error of the regression slopes and one standard error of the $NO_z$ means, respectively. Mean $NO_z$ measurement uncertainty is estimated as 0.2 ppbv (1 sigma).