# Peer review of "Ozone Response to Emission Reductions in the Southeastern United States"

_Atmospheric Chemistry and Physics, 2017_

## Referee Comment (RC1) · Anonymous Referee #1 · 1 Sep 2017

Review of "Seasonal Changes in Ozone and its Production in the Southeastern United States between 1992 and 2014" by Charles L. Blanchard and George M. Hidy Blanchard and Hidy use long term SEARCH network data to look at how ozone is responding to emission changes in the Southeastern US, and relate this to the concept of ozone production efficiency (OPE). They find ozone is decreasing in response to NOx controls while OPE is increasing. The increase in OPE indicates an increasing effectiveness in NOx controls. The article is informative, particularly for those looking at trends in the southeastern US, and has enough discussion of the system to be of interest to others. However, there are some current issues with the manuscript that should be addressed before publication. Their way of measuring OPE can present biases, some

of which they capture. First, if not all of the NOz is measured, that will lead to a high bias. Second, they present their method for trying to make sure that the background ozone is not biasing the calculation. While I appreciate the effort, they really don't show that it works. (They do an analysis, but in the end, it is not very satisfying and needs a bit more analysis and justification.) The comparison of their OPE's to modeled values is of interest, but again, unsatisfying. Do the Liu et al., values of up to 80 make sense? Do OPEs of 20 for a NOx of 1 ppb make sense given their results? It would be good if they provide some critical analysis. If the OPEs increased from their values of, currently, about 20, to 80, while the NOz decreases from 1 to 0.1 ppb. Wouldn't this lead to ozone levels below background and well below their asymptotic values? Please comment. When they say that for a limit of OPE approaching zero... Why does one presuppose such a limit? That is in contrast to Liu et al. It is not apparent they are capturing all of the oxidized N in their work. How much of the organic N is measured (e.g., the fraction with the PM)? When they are using NOz, are they missing much (how much)?

Discussion of VOC reactivity: OH reactivity is a poorly used measure of ozone formation from VOCs. USE MIR or MOIR ([Carter, 1994]) Figure 2: What is the -10th %ile? Do you mean 10th %ile? (No minus) The Abstract is currently not very informative. More hard results should be provided. To say "O3 declines are less than proportional to the decreases in NOx" is obvious to most folks... there is a very non-zero ozone background, so you expect less than proportional. While they say OPE has increased, they don't say by how much. They don't say what are the ozone reductions. Provide some details. If I just read the abstract I would not have learned much, and would not really be included to read the article. The atmospheric chemistry primer (section 2.1) is too basic for the readers of ACPD. Some parts are fine but assume the readers know reactions R1-R7.

In summary, the paper is informative, though I believe a number of modifications and further analysis are required for acceptance.

Carter, W. P. L. (1994), Development of Ozone Reactivity Scales for Volatile Organic-Compounds, J. Air Waste Manage. Assoc., 44(7), 881-899.

---

## Short Comment (SC1) · 13 Sep 2017

Some findings from the following paper are highly relevant to discussions here.

Lin, M.Y., W. Horowitz, R. Payton, A.M. Fiore, G. Tonnesen (2017). US surface ozone trends and extremes from 1980 to 2014: Quantifying the roles of rising Asian emissions, domestic controls, wildfires, and climate. Atmos. Chem. Phys., doi:10.5194/acp-17-2943-2017.

This paper shows that the O3 decreases driven by NOx controls were most pronounced in the southeastern US, where the seasonal onset of biogenic isoprene emissions and

[Figure]

NOx-sensitive O3 production occurs earlier than in the northeast.

---

## Author Comment (AC1) · 3 Oct 2017

We thank the reviewer for helpful suggestions and address the reviewer's questions in the following paragraphs.

The SEARCH measurements of NOy were designed to capture particulate nitrate and organic nitrates, as well as NO, NO2, HNO3, and other oxidized nitrogen species. The NOy sampler derives from the ESE instrument discussed in Williams et al. (1998), which was one of five instruments for which measurements of NOy reproduced the sum of separately measured NOy species. Additional testing in 2013 showed that SEARCH NOy measurements agreed with the sum of measured mixing ratios of NO,

[Figure]

NO2, HNO3, particulate nitrate, alkyl nitrates, and peroxy–alkyl nitrates (Hidy et al., 2014). The NOy measurements have therefore been shown to capture oxidized nitrogen species near both the beginning and the end of the study years. As described in the manuscript, the method for measuring NO2 is NO2-specific, but over time the instruments utilized three different types of lamps for the photolytic conversion of NO2 to NO. We therefore tested for biases in the O3-NOz relationships by determining O3-HNO3 relationships as described, and concluded that similar temporal changes occurred in both sets of relationships.

We concluded that our observed increases in the slopes of summer O3 versus NOz did not result from variations in background O3 based on a set of auxiliary analyses that we provided. Rather than generating more auxiliary analyses, we argue here that other studies sufficiently support and justify our conclusions. O3 decreases driven by reductions of NOx emissions between 1980 and 2014 were most pronounced in the southeastern US, where the seasonal onset of biogenic isoprene emissions and NOx-sensitive O3 production occurs earlier than in the northeastern U.S. (Lin et al., 2017). Lin et al. (2017) show that rising NOx emissions in Asia have increased modeled North American background O3 levels (based on model simulations with zero North American emissions) by ∼0.2 ppbv yr-1 in the southeastern U.S. in summer. The model-predicted increase in background O3 in the southeastern U.S. is too small to be a systematic cause of our observed twofold (or more) increase in the slopes of summer O3 versus NOz. Moreover, the actual O3 levels occurring in the southeastern U.S. during our study period would have been influenced by transport of air masses affected by non-zero North American emissions occurring upwind of our study area, i.e., by regional background, whose changes likely differ from changes in North American background.

Observed trends in the 5th percentile O3, have previously been used as indicators of changes in regional or continental background O3 (e.g., Wilson et al., 2012). The 5th percentile peak daily 8-hour O3 mixing ratios decreased during summer at rural sites throughout the southeastern U.S. between 1988 and 2014 (Lin et al., 2017). By this

measure, background O3 levels were not increasing in the southeastern U.S. during our study period and therefore could not have introduced a positive bias in observed ozone production efficiency (OPE, estimated as the slopes of summer O3 versus NOz). The trend in the 95th percentile summer peak daily 8-hour O3 mixing ratios in the southeastern US reported by Lin et al. is $\sim$ -0.8 to -1.8 ppbv yr-1, with downward trends occurring in other seasons as well. Our findings are comparable: between 1999 and 2014, the highest peak daily 8-hour O3 mixing ratios occurring each month) declined at all SEARCH sites at statistically significant (p < 0.01) rates averaging $\sim$1 ppbv yr-1 (our Figure 3).

Anthropogenic emissions and long-range transport (long-range tropospheric + stratospheric) O3 each accounted for about 40% (15 – 20 ppbv) of model-predicted O3 below 1 km altitude at Huntsville, AL, during June 2013, while long-range transport accounted for $\sim$80% of model-predicted O3 above 4 km altitude (Johnson et al., 2016). Using ozonesondes that are launched on a typically weekly schedule, vertical O3 mixing ratio profiles have been determined by the University of Alabama in Huntsville, Alabama, since 1999 (Newchurch et al., 2003; Johnson et al., 2016; University of Alabama, 2017; NOAA, 2017). We obtained the ozonesonde data (University of Alabama, 2017; NOAA, 2017) and identified the following statistically significant trends in the lower layers that are relatively more influenced by local and regional emissions according to Johnson et al. (2016): -0.25 $\pm$ 0.11 ppbv y-1 (p < 0.05) at 0.5 km, -0.40 $\pm$ 0.10 ppbv y-1 (p < 0.0001) at 1 km, -0.42 $\pm$ 0.09 ppbv y-1 (p < 0.0001) at 2 km, and -0.57 $\pm$ 0.13 ppbv y-1 in monthly averages of O3 measurements made throughout the interval 1 – 2 km (p < 0.001). At higher altitudes where Johnson et al. (2016) predicted that long-range transport is the dominant source of O3, no trends occurred: 0.06 $\pm$ 0.08 ppbv y-1 (p > 0.1) at 4 km and 0.09 $\pm$ 0.19 ppbv y-1 (p > 0.1) at 8 km. The Huntsville ozonesonde data support our conclusion that changes in observed OPE are not biased by trends in transport of background O3.

We suggest revising the second paragraph of Section 4.4.2 as follows: "The increase

in OPE with decreasing ambient NOx and NOz is also consistent with computations by Liu et al. (1987), which show relatively constant summer OPE of $\sim$ 7 - 10 for ambient NOx exceeding $\sim$7 ppbv and increases in OPE to $\sim$20 as NOx declines from $\sim$7 to $\sim$1 ppbv. While the numerical results of the modeling calculations by Liu et al. (1987) are specific to the modeled conditions, increasing OPE results from multiple factors that are pertinent to other conditions, such as radical reactions involving VOCs and NOx (Lin et al., 1988)." This revision eliminates reference to model-predicted OPE at lower NOx mixing ratios. The SEARCH data do not provide an observational test of Liu et al.'s (1987) model prediction that OPE increases to $\sim$60 - 80 as NOx declines to $\sim$0.1 ppbv to $\sim$0.01 ppbv, because fewer than 0.2% of the 2 p.m. SEARCH NOx mixing ratios were below 0.1 ppbv.

The first paragraph of Section 4.4.3 can be shortened to read: "Where NOx limits re-action rates, O3 production is the product of OPE and ambient NOx mixing ratios as determined for specific ambient conditions (Liu et al., 1987), so O3 reductions depend on changes in both OPE and NOx. At present, there is no clear indication from the SEARCH data that OPE will continue to increase, level off, or begin decreasing. The post-1990s O3 trend provides one guide to future average rates of O3 reduction in the sense that the rates of O3 reduction during the next decade are unlikely to deviate dra-matically from those of the recent past. This result would be expected if OPE remains roughly constant or decreases somewhat, as indicated in Figure 10 for the years since about 2009. Previous work indicates that VOC reactivity and O3 losses contribute to nonlinearity; at ambient NOx mixing ratios less than $\sim$0.4 ppbv, O3 loss suppresses OPE, and below $\sim$80 pptv NOx, OPE becomes negative (Lin et al., 1988)."

The importance of isoprene emissions for ozone production in the southeastern U.S. is well established (e.g., Chameides et al., 1988; Chameides and Cowling, 1995; Frost et al., 1998; Starn et al., 1998; Wiedinmyer et al., 2006; Zhang et al., 2014; Lin et al., 2017) and requires no further analysis. In our view, the suggested evaluation of reac-tivity utilizing the Carter (1994) MIR and MOIR O3 reactivity scales would be limited by

the availability of data, would not provide new insight, and would not contribute to our assessment of the relationship between O3 and NOx in the southeastern U.S. Panel (d) of Figure 6, intended to be illustrative, can be eliminated if it is a distraction. We can simply reference the literature on isoprene reactivity and the significance of seasonal isoprene emissions in the Southeast.

We suggest rewording the second half of the abstract as follows: "Between 1996 and 2015, annual 4th-highest daily peak 8-hour O3 mixing ratios at EPA monitoring sites in Georgia, Alabama, and Mississippi exhibited statistically-significant ($p < 0.0001$) linear correlations with annual NOx emissions in those states, decreased by $\sim$30% on average, and declined at rates averaging $\sim$1 ppbv y-1. Ozone production efficiency (OPE, molecules of O3 produced per molecule of NOx oxidized) increased by factors of $\sim$2 or more between 1999 and 2014, which partially offset the $\sim$60% NOx emission reductions and limited the O3 decreases. The results suggest increasing responsiveness of O3 to NOx as NOx emissions decline, but the effectiveness of ongoing NOx emission reductions will depend on the balance between changes in observed OPE and ambient NOx in the context of changes in anthropogenic emissions of volatile organic compounds (VOC)."

The correct statement in the caption of Figure 2 should be "a) trends (ranges denote 90th and 10th percentile site's values)."

We will shorten Section 2.1 (background information on atmospheric chemistry), relying on textbook and research reviews for details. We agree that this material appears extensively in the literature and does not need to be expanded in our manuscript, which is not intended as a comprehensive review.

References Carter, W. P. L.: Development of ozone reactivity scales for volatile organic compounds, J. Air Waste Manage. Assoc., 44(7), 881-899, 1994.

Chameides, W., Lindsay, R., Richardson, J., and Kiang, C.: The role of biogenic hydrocarbons in urban photochemical smog: Atlanta as a case study, Science, 24, 1473–

1475, 1988.

Chameides, W. and E. Cowling: The State of the Southern Oxidants Study: Policy Relevant Findings in Ozone Pollution Research, 1988-1994, Southern Oxidant Study, College of Forest Resources, North Carolina State University, Raleigh, NC, 1995.

Frost, G. J., M. Trainer, G. Allwine, M. P. Buhr, J. G. Calvert, C. A. Cantrell, F. C. Fehsenfeld, P. D. Goldan, J. Herwehe, G. Hubler, W. C. Kuster, R. Martin, R. T. McMillen, S. A. Montzka, R. B. Norton, D. D. Parrish, B. A. Ridley, R. E. Shetter, J. G. Walega, B. A. Watkins, H. H. Westberg, and E. J. Williams: Photochemical ozone production in the rural southeastern United States during the 1990 Rural Oxidants in the Southern Environment (ROSE) program, J. Geophys. Res. Atmos., 103(D17), 22491-22508, 1998.

Hidy, G., C. Blanchard, K. Baumann, E. Edgerton, S. Tanenbaum, S. Shaw, E. Knipping, I. Tombach, J. Jansen and J. Walters: Chemical climatology of the southeastern United States, 1999-2013, Atmos. Chem. Phys., 14, 11893-11914, 2014.

Johnson, M., S. Kuang, L. Wang, and M. Newchurch: Evaluating summer-time ozone enhancement events in the southeast United States, Atmosphere, 7, 108, doi:10.3390/atmos7080108, 2016.

Lin, M., L. W. Horowitz, R. Payton, A. M. Fiore, and G. Tonnesen: U.S. surface ozone trends and extremes from 1980 to 2014:quantifying the roles of rising Asian emissions, domestic controls, wildfires, and climate. Atmos. Chem. Phys., 17, 2943–2970, doi:10.5194/acp-17-2943-2017, 2017.

Lin, X., M. Trainer, and S. C. Liu: On the nonlinearity of the tropospheric ozone production., J. Geophys. Res. Atmos., 93(D12), 15879 – 15888, 1988.

Liu, S. C., M. Trainer, F. C. Fehsenfeld, D. D. Parrish, E. J. Williams, D. W. Fahey, G. Hubler, and P. C. Murphy: Ozone production in the rural troposphere and the implications for regional and global ozone distributions, J. Geophys. Res., 92(D4), 4191-4207,
1987.

National Oceanic and Atmospheric Administration (NOAA): ESRL/GMD FTP Data Finder, ftp://aftp.cmdl.noaa.gov/data/ozwv/Ozonesonde/Huntsville,%20Alabama/100%20Meter%20Average%20Files/ html (last access October 2, 2017), 2017.

Newchurch, M., M. Ayoub, S. Oltmans, B. Johnson, and F. Schmidlin: Vertical distributions of ozone at four sites in the United States. J. Geophys. Res. 208 (D1), 4031, doi:10.1029/2002JD002059, 2003.

Starn, T. K., P. B. Shepson, S. B. Bertman, J. S. White, B. G. Splawn, D. D. Riemer, R. G. Zika, K. Olszyna: Observations of isoprene chemistry and its role in ozone production at a semirural site during the 1995 Southern Oxidants Study, J. Geophys. Res., 103, 22425–22435, doi: 1 0.1029/98JD01279, 1998. University of Alabama Huntsville (UAH). 2017. Huntsville ozonesonde station. http://www.nsstc.uah.edu/atmchem/about_ozonesonde.html (last access October 2, 2017), 2017.

Wiedinmyer, C., X. Tie, A. Guenther, R. Neilson, and C. Granier: Future changes in biogenic isoprene emissions: how might they affect regional and global atmospheric chemistry?, Earth Interactions, 10, https://doi.org/10.1175/EI174.1, 2006.

Williams, E. J., K. Baumann, J. M. Roberts, S. B. Bertman, R. B. Norton, F. C. Fehsenfeld, S. R. Springston, L. J. Nunnermacker, L. Newman, K. Olszyna, J. Meagher, B. Hartsell, E. Edgerton, J. R. Pearson, and M. O. Rodgers: Intercomparison of ground-based NOy measurement techniques, J. Geophys. Res. 103 (D17), 22261–22280, 1998.

Wilson, R. C., Z. L. Fleming, P. S. Monks, G. Clain, S. Henne, I. B. Konovalov, S. Szopa, and L. Menut: Have primary emission reduction measures reduced ozone across Europe? An analysis of European rural background ozone trends 1996–2005, Atmos. Chem. Phys., 12, 437–454, doi:10.5194/acp-12-437-2012, 2012.
Zhang, Y., W. Wang, S.-Y. Wu, K. Wang, H. Manoura, and Z. Wang: Impacts of updated emission inventories on source apportionment of fine particles and ozone over the southeastern U.S., Atmos. Environ., doi: 10.1016/j.atmosenv.2014.01.035, 2014.

---

## Author Comment (AC2) · 3 Oct 2017

We appreciate this reference as several of its results are highly relevant. We have drawn on this citation in addressing some of the questions raised in the formal review.

---

## Referee Comment (RC2) · D. Parrish (Referee) · 12 Dec 2017

Review of " Ozone Response to Emission Reductions in the Southeastern United States" by Charles L. Blanchard and George M. Hidy

MS Number: acp-2017-534

**Summary:**

The paper presents some useful analysis of a very valuable data set. The SEARCH data cover more than 2 decades of measurements made at eight sites in the southeastern U.S., and certainly deserve careful analyses from many perspectives. Some of that analysis is presented in this paper, but major portions of the analysis are incorrect, and are consequently misleading. Major issues that require attention are detailed below, followed by listing of more minor issues with suggestions for improvement. I suggest that this paper not be accepted before it has been extensively revised to address the issues detailed below.

**Major issues:**

1) In the abstract the authors conclude that "The $O_3$ declines are less than proportional to the decreases in NOx emissions: emissions decreased by ~60% and $O_3$ maxima declined ~30 – 35% at rates averaging ~1 ppbv y-$^1$." However, the authors neglect to consider the contribution of transported background $O_3$ contributions to the $O_3$ maxima. When this contribution is properly considered, the declines will be much more nearly proportional (see comment 4) below for more details).

2) In the abstract the authors also conclude that "Ozone production efficiency (OPE, molecules of $O_3$ produced per molecule of NOx oxidized) increased between 1999 and 2014, which affected the magnitude of the $O_3$ response to NOx emission reductions by partially offsetting precursor decreases and contributing to the nonlinear $O_3$ response." However, the OPE analysis presented is flawed (see comment 7) below for more details), and this conclusion is simply not correct. It must be removed.

3) The abstract ends with the conclusion that "The results suggest increasing responsiveness of $O_3$ to NOx, but the effectiveness of ongoing NOx emission reductions will depend on the balance between changes in observed OPE and ambient NOx in the context of changes in anthropogenic emissions of volatile organic compounds (VOC). This conclusion is not supported by valid analysis in this paper; it must also be removed.

4) On pg. 8 the authors make two observations with regard to figure 2. First, "$O_3$ mixing ratios are declining toward nonzero values, as indicated by the statistically-significant (p < 0.0001) intercepts of ~45 – 50 ppbv." Second, "the $O_3$ declines are less than proportional to the decreases in NOx emissions, as indicated by the ~60% emission reduction and ~30 – 35% $O_3$ declines shown in Figure 2, about equivalent to the national trends discussed in Section 2.2". These two observations are closely connected and should be discussed further.

First, the intercepts can be reasonably interpreted as U.S. background $O_3$ contributions (i.e., the $O_3$ concentrations that would be present in the absence of U.S. anthropogenic precursor emissions) to these $O_3$ concentrations. The derived intercepts of ~45 – 50 ppbv can be compared to other estimates of U.S. background $O_3$ concentrations. *Berlin et al.* [2013] estimate mean regional background $O_3$ concentrations of 48 ppbv to 59 ppbv on exceedance days in the Houston TX area. However, these estimates include $O_3$ contributions from transport to the area from other regions of the U.S., and thus are higher than true U.S. background $O_3$ concentrations. It should also be noted that these estimates for Houston

exceedance days are higher than the regional average of all summer days. *Parrish et al.* [2017a] note that the highest ozone design values (i.e., the 3 year running mean of the 4th highest 8-hour average $O_3$ concentration) in Southern California air basins are converging toward of limit of $62.0 \pm 1.9$ ppb, which they identify as the ozone design values that would result from only U.S. background ozone concentrations. The California background ozone concentrations are higher than in Texas or the Southeastern United States discussed in the present paper due to differences in state orography, site altitudes and proximity to major areas of surface impact from stratospheric intrusions. Such comparisons should be discussed in the present paper.

Second, it would be more informative to compare the percentage declines in NOx emissions to the percentage declines in $O_3$ after subtracting the intercepts; such a comparison would give significantly larger relative $O_3$ reductions, and these higher results would be closer in magnitude to the relative reductions in NOx emissions; this comparison would more faithfully reflect the reduction in the anthropogenic contribution to observed $O_3$ concentrations. For example, *Parrish et al.* [2017a] find that the ozone enhancement above background in Southern California has decreased with an e-folding time of 21.9 years, which corresponds to a decrease of 4.5%/yr, larger than the value of 2.8%/yr given by *Pollack et al.* [2013] as cited by the authors. This difference arises because *Pollack et al.* [2013] did not subtract the background before deriving the relative rate of decrease.

Considering $O_3$ trends after background subtraction makes a substantial difference. In Southern California this approach implies that the anthropogenic enhancement of ozone (the only pollution contribution that is within the control of U.S. policy makers) has decreased by a factor of 5 from 1980 to 2015. This factor is larger than generally appreciated, and is an important success story for air quality improvement efforts in the U.S. that deserves wider recognition. It is also notable that this rate of decrease is between the rates of decrease of ambient VOCs and NOx (7.3% yr-1 and 2.6% yr-1, respectively, 1960 – 2010) in Southern California, as cited by the present authors. This same consideration of the change in the anthropogenic enhancement of ozone should be presented in this paper for the Southeastern U.S. I realize that the references cited in Table S2 did not subtract the U.S. background concentration before calculating the tabulated relative ozone decreases; this likely explains much of the regional difference between Southern California and the Southeastern U.S. I strongly recommend that this subtraction be done and discussed in this paper.

5) The sentence beginning on Pg. 9, line 2 ("Both EPA (Figure 2) and SEARCH (Figure 3) data suggest that $O_3$ mixing ratios increased during the 1990s, then began declining.") suggests that the trends in Figure 3 should be calculated only after the increase had ended, i.e., beginning in the year ~2000. When this is done, some of the trends (i.e., CTR, YRK and OAK) will be steeper, and there may be better agreement among the trends at the different sites.

6) The correlations shown in Figure 5 are misleading, and this figure should not be included without extensive modification. One major problem is that the figure combines wintertime data, when $O_3$ concentrations may be reduced below those in transported background air due to titration by NO emissions, with summertime data, when $O_3$ concentrations are increased above those in transported background air due to photochemical $O_3$ production. The figure should either include data from one season only, or plot Ox (= $O_3$ + $NO_2$) concentrations, which are much less sensitive to the NO titration, instead of $O_3$ only concentrations. The

SEARCH data are somewhat unique in having simultaneous high quality $O_3$ and $NO_2$ data, and this analysis should take advantage of this uniqueness. This plot may be further confused by wintertime conversion of NOx to NOz through $NO_3$ and $N_2O_5$ chemistry, which destroys rather than produces $O_3$.

7) Section 4.4 attempts to quantify ozone production efficiency (OPE) from observations, but this entire discussion must be rethought. There may be something of value in the extensive analysis that the authors performed, but the current discussion is simply not correct. Specific difficulties include:

- Ozone is quite low (≤20 ppb) at low NOz concentrations in figures 8 and 9; this immediately identifies a clear problem in the analysis. The observationally based determination of OPE implicitly assumes that "background air" contains zero NOz concentrations and $O_3$ concentrations representing regional background transported into the region. Variations of $O_3$ concentrations transported into the region must be negligible compared to the $O_3$ produced within the region or locally. That is simply not the case here. With few exceptions, all of the $O_3$ concentrations in Figure 9 are <65 ppb. *Berlin et al.* [2013] show that regional background $O_3$ concentrations varied between ~10 and 70 ppb in the Houston area in the mid 2000s. Thus, it is conceivable that Figure 8 and 9 (particularly the latter) are dominated by $O_3$-NOz relationships in the transported regional background, and provide little or no information regarding ozone formation within the SEARCH region.

- Figure 9 gives linear fits of observed $O_3$ vs NOz for one year, and Table S4 gives the results for all years of data. The figure below shows the relationship between the derived slopes and intercepts for all years and all sites in Table S4. If the slopes were indeed providing information about the local and regional photochemistry, they would be expected to be independent of the intercepts, which reflect the regional background; such independence is clearly not seen. For the two urban sites (BHM and JST) the intercepts account for almost 80 of the variability in the slope.

- The paragraph beginning on pg. 12, line 7 attempts to account for the influence of depositional loss of NOz on derived OPE values, and the influence of varying background $O_3$ concentrations. Unfortunately, the three different methods employed, yield quite different OPE values (Figures S15 – 17). Also, the results do not make good physical sense; e.g. how can OPE be near zero in 2001 at JST? Thus, this discussion increases the skepticism with which the entire analysis must be considered.

[Figure]

- The paragraph beginning on pg. 13, line 4 compares the intercepts of year-specific regressions for 2013 (~20 ppb $O_3$) with other estimates of background levels. However, this comparison is not valid. Some of the references cited (Lefohn et al., 2014; Dolwick et al., 2015) are modeling studies that discuss U.S. background $O_3$ according to the EPA definition, which is the $O_3$ concentration that would exist if all U.S. anthropogenic emissions of ozone precursors were reduced to zero. Others (Chan and Vet, 2010) report observationally-based estimated baseline $O_3$ concentrations in the absence of any continental influences. These two concepts are very different from regional background $O_3$, i.e. the $O_3$ concentration actually transported into the region of interest, including from other U.S. regions that are rich in anthropogenic emissions of ozone precursors. A comparison with the work of *Berlin et al.* [2013] is much more appropriate for discussion of the SEARCH region.

- In Section 4.4.2 the authors compare their results with cited work from the published literature. Many of the references cited give results from studies that suffer from the same problems as plague the present work. For example *Travis et al.* (2016) follow much the same approach as the present paper - they interpret the slope of the correlations of Ox vs. NOz as OPE with no analysis to ensure that the low Ox-low NOz air and the high Ox-high Oz air actually represent similar background Ox and NOz concentrations, to which varying amounts of precursors were injected and subsequently photochemically processed. Reliable analysis of OPEs requires careful plume analysis, similar to that presented in *Neuman et al.*, 2009 (a reference that is not cited in the present paper). One approach to deriving OPEs from surface site data is given by *McDuffie et al.*, 2009 (a reference that the authors cite, but do not discuss the OPE results therein.) The references to *Liu et al.* (1987) and *Lin et al.* (1988) are not germane to the present discussion, as these results are from a very early global model, and report the total ozone produced when all VOCs, including only relatively unreactive VOCs are completely oxidized over months.

- Finally, a very simple argument makes it quite clear that something is amiss in the entire OPE analysis. Section 4 begins with a discussion of trends in NOx emissions, emphasizing a reduction of a factor of ~3 between 1996 and 2014. Figure 10 suggests that OPE has increased by a factor of ~5. If both of these findings were correct, then $O_3$ concentrations, at least from local and regional production, would have increased, not decreased, over this period. Yet the authors note that $O_3$ concentrations have in fact decreased. There is a critical inconsistency buried in this analysis

- Section 4.4.3 is highly speculative, and based upon inaccurate OPEs as discussed above. It should be eliminated in its entirety, or at least extensively modified if the issues listed above can be effectively addressed.

7) The Conclusions section must be revised consistent with the revisions needed to address the above issues.

**Minor issues:**

1) Line 11: typo - "... in in Alabama and Georgia."

2) In my opinion Figure 1a would be more informative as a semi-log plot. Then the NOx emission and nitrate deposition traces would parallel each other, and the linear slope of the

log-transformed data would be directly proportional to the % decrease/yr.  If the NOx emissions were plotted on the right axis and the deposition data on the left with the same factor change on each axis, but the offset on each axis chosen properly, then the emissions and deposition curves would be approximately superimposed.

3) Pg. 8, lines 18-19: At least the SEARCH downward trends in mean annual $HNO_3$ concentrations in %/yr that can be derived from Figure S4 should be compared to the corresponding trends in NOx emission and nitrate deposition.  (The EPA $HNO_3$ trends do not seem to make good physical sense.)  Figure S4 also would be more informative as a semi-log plot.

4) The de Gouw et al., 2014 reference is omitted from the References list.

5) I do not understand the sentence beginning on Pg. 8, line 23: "Spatial variability of the annual 4th-highest daily peak 8-hour $O_3$ mixing ratios has decreased (Figure 2), consistent with an analysis of data from a larger number of U.S. and European locations (Paoletti, et al., 2014)."  Figure 2 has no direct information regarding spatial variability.  It is true that the spread in the percentiles of the 4th highest $O_3$ concentrations has decreased, but his is only to be expected as the absolute magnitude of the anthropogenic ozone enhancement has decreased.  In terms of absolute ozone concentration, then the spatial variability is expected to have decreased simply because all of the region is approaching the U.S. background $O_3$ concentration, which is expected to have small spatial variability in the Southeastern U.S. This sentence should be more clearly explained.

6) I suggest that the sentence beginning on Pg. 9, line 7 be reworded: "The meteorological factors having the strongest influence on daily peak 8-hour $O_3$ mixing ratios at SEARCH sites are daily maximum temperature and mid-day relative humidity (RH), whose variations cause daily peak 8-hour $O_3$ mixing ratios to vary by ~ ±30 percent from mean peak 8-hour O3 mixing ratios (Blanchard et al., 2014)."  I assume that these results are simply correlations, without proof of cause; thus the sentence should read something like: "The meteorological factors correlating most strongly with daily peak 8-hour $O_3$ mixing ratios at SEARCH sites are daily maximum temperature and mid-day relative humidity (RH), with variations of daily peak 8-hour $O_3$ of ~ ±30 percent from mean peak 8-hour $O_3$ mixing ratios (Blanchard et al., 2014)."

7) The sentence beginning on Pg. 9, line 24 is likely misleading: "Background $O_3$ may also represent an increasing absolute contribution in our study area, as multiple studies have demonstrated increasing trends in global background $O_3$ mixing ratios."  The cited studies have all focused on northern mid-latitudes, where the background $O_3$ mixing ratios have indeed increased.  However, *Parrish et al.* [2017b] show that increase generally ended in the early to mid 2000s.  Further, *Berlin et al.* [2013] show that baseline ozone concentrations in air flowing into Texas from the Gulf of Mexico have not changed significantly over the 1990-2010 period.  It is likely that the Gulf of Mexico inflow better represents the background ozone affecting the Southeastern U.S., which is the subject of this paper.

8) The sentence on Pg. 10, lines 13-16 clearly refers to data over the full year.  It would be more informative to include the % of the VOC reactivity due to isoprene just for the summer months when both the high isoprene and high ozone concentrations occur.  Similarly, the alkene and aromatic contributions to average VOC OH reactivity for the high ozone summer season should be contrasted with the annual average numbers that are given.

**References**

Berlin, S.R., A.O. Langford, M. Estes, M. Dong, and D.D. Parrish (2013), Magnitude, decadal changes and impact of regional background ozone transported into the greater Houston, Texas area, *Environ. Sci. Technol*, *47*(24), 13985-13992, doi:10.1021/es4037644.

Neuman, J. A., et al. (2009), Relationship between photochemical ozone production and NOx oxidation in Houston, Texas, *J. Geophys. Res., 114*, D00F08, doi:10.1029/2008JD011688.

Parrish, D. D., Young, L. M., Newman, M. H., Aikin, K. C., & Ryerson, T. B. (2017a). Ozone design values in Southern California's air basins: Temporal evolution and U.S. background contribution. Journal of Geophysical Research: Atmospheres, 122. https://doi.org/10.1002/2016JD026329

Parrish, D. D., Petropavlovskikh, I., & Oltmans, S. J. (2017b). Reversal of long-term trend in baseline ozone concentrations at the North American West Coast. Geophysical Research Letters, 44. bhttps://doi.org/10.1002/2017GL074960

---

## Author Comment (AC3) · 9 Jan 2018

We appreciate the reviewer's affirmation of the value of the SEARCH program and database. In summary, the ∼20-year SEARCH measurement record is not in dispute. Moreover, the primary result that we present, which is a quantification of the responses of ozone (O3), nitrogen oxides (NOx), and NOx reaction products to changes in anthropogenic NOx emissions in the southeastern U.S., is not in dispute either. We have documented observed trends and relationships, and we have now expanded these descriptions in accordance with the reviewer's suggestions. We have also revised the manuscript following each of the reviewer's major and minor points.

---

## Author Response (AR1)

Response to review

We appreciate the reviewer's affirmation of the value of the SEARCH program and database. In summary, the ~20-year SEARCH measurement record is not in dispute. Moreover, the primary result that we present, which is a quantification of the responses of ozone ($O_3$), nitrogen oxides ($NO_x$), and $NO_x$ reaction products to changes in anthropogenic $NO_x$ emissions in the southeastern U.S., is not in dispute either. We have documented observed trends and relationships, and we have now expanded these descriptions in accordance with the reviewer's suggestions. In addition, the reviewer criticizes our interpretations of underlying causes, and, more specifically, identifies portions of the analysis that the reviewer considers incorrect. We have therefore revised the manuscript as summarized following each of the reviewer's major and minor points. In our revision, we identify factual results that are not in dispute, distinguish undisputed results from questions of interpretation, and provide additional analyses to further clarify our interpretations of $O_3$ changes over time. For convenience, the entire text of the reviewer is reproduced here in red, interspersed with our responses.

**Summary**:

The paper presents some useful analysis of a very valuable data set. The SEARCH data cover more than 2 decades of measurements made at eight sites in the southeastern U.S., and certainly deserve careful analyses from many perspectives. Some of that analysis is presented in this paper, but major portions of the analysis are incorrect, and are consequently misleading. Major issues that require attention are detailed below, followed by listing of more minor issues with suggestions for improvement. I suggest that this paper not be accepted before it has been extensively revised to address the issues detailed below.

**Major issues:**

1) In the abstract the authors conclude that "The O3 declines are less than proportional to the decreases in NOx emissions: emissions decreased by ~60% and O3 maxima declined ~30 – 35% at rates averaging ~1 ppbv y-1." However, the authors neglect to consider the contribution of transported background O3

contributions to the O3 maxima. When this contribution is properly considered, the declines will be much more nearly proportional (see comment 4) below for more details).

2) In the abstract the authors also conclude that "Ozone production efficiency (OPE, molecules of O3 produced per molecule of NOx oxidized) increased between 1999 and 2014, which affected the magnitude of the O3 response to NOx emission reductions by partially offsetting precursor decreases and contributing to the nonlinear O3 response." However, the OPE analysis presented is flawed (see comment 7) below for more details), and this conclusion is simply not correct. It must be removed.

3) The abstract ends with the conclusion that "The results suggest increasing responsiveness of O3 to NOx, but the effectiveness of ongoing NOx emission reductions will depend on the balance between changes in observed OPE and ambient NOx in the context of changes in anthropogenic emissions of volatile organic compounds (VOC). This conclusion is not supported by valid analysis in this paper; it must also be removed.

We revised the abstract to address points raised by both reviewers. The preceding three comments are discussed below (reviewer's point 1 is addressed under point 4 and reviewer's point 2 under point 7). Regarding point 1, our results can simply be presented as rates over time with stated error limits (as we have done), without asserting that they are either proportional or non-proportional. Regarding reviewer's point 3, the last sentence of the abstract was intended as a caveat about the limitations of extrapolating past to future trends, rather than a conclusion, and has been rephrased.

4) On pg. 8 the authors make two observations with regard to figure 2. First, "O3 mixing ratios are declining toward nonzero values, as indicated by the statistically-significant ($p < 0.0001$) intercepts of ~45 – 50 ppbv." Second, "the O3 declines are less than proportional to the decreases in NOx emissions, as indicated by the ~60% emission reduction and ~30 – 35% O3 declines shown in Figure 2, about equivalent to the national trends discussed in Section 2.2". These two observations are closely connected and should be discussed further. First, the intercepts can be reasonably interpreted as U.S. background O3 contributions (i.e., the O3 concentrations that would be present in the absence of U.S. anthropogenic precursor emissions) to these O3 concentrations. The derived intercepts of ~45 – 50 ppbv can be compared to other estimates of U.S. background O3 concentrations. Berlin et al. [2013] estimate mean regional

background O3 concentrations of 48 ppbv to 59 ppbv on exceedance days in the Houston TX area. However, these estimates include O3 contributions from transport to the area from other regions of the U.S., and thus are higher than true U.S. background O3 concentrations. It should also be noted that these estimates for Houston exceedance days are higher than the regional average of all summer days. Parrish et al. [2017a] note that the highest ozone design values (i.e., the 3 year running mean of the 4th highest 8-hour average O3 concentration) in Southern California air basins are converging toward of limit of 62.0 ± 1.9 ppb, which they identify as the ozone design values that would result from only U.S. background ozone concentrations. The California background ozone concentrations are higher than in Texas or the Southeastern United States discussed in the present paper due to differences in state orography, site altitudes and proximity to major areas of surface impact from stratospheric intrusions. Such comparisons should be discussed in the present paper.

We added comparisons to other studies and discussed them.

Second, it would be more informative to compare the percentage declines in NOx emissions to the percentage declines in O3 after subtracting the intercepts; such a comparison would give significantly larger relative O3 reductions, and these higher results would be closer in magnitude to the relative reductions in NOx emissions; this comparison would more faithfully reflect the reduction in the anthropogenic contribution to observed O3 concentrations. For example, Parrish et al. [2017a] find that the ozone enhancement above background in Southern California has decreased with an e-folding time of 21.9 years, which corresponds to a decrease of 4.5%/yr, larger than the value of 2.8%/yr given by Pollack et al. [2013] as cited by the authors. This difference arises because Pollack et al. [2013] did not subtract the background before deriving the relative rate of decrease. Considering O3 trends after background subtraction makes a substantial difference. In Southern California this approach implies that the anthropogenic enhancement of ozone (the only pollution contribution that is within the control of U.S. policy makers) has decreased by a factor of 5 from 1980 to 2015. This factor is larger than generally appreciated, and is an important success story for air quality improvement efforts in the U.S. that deserves wider recognition. It is also notable that this rate of decrease is between the rates of decrease of ambient VOCs and NOx (7.3% yr-1 and 2.6% yr-1, respectively, 1960 – 2010) in Southern California, as cited by

the present authors. This same consideration of the change in the anthropogenic enhancement of ozone should be presented in this paper for the Southeastern U.S. I realize that the references cited in Table S2 did not subtract the U.S. background concentration before calculating the tabulated relative ozone decreases; this likely explains much of the regional difference between Southern California and the Southeastern U.S. I strongly recommend that this subtraction be done and discussed in this paper.

We added intercept-corrected comparisons for the EPA AL-GA data in the discussions of Figure 2 and the SEARCH data in Table S2. We did not do this for papers cited in Table S2.

5) The sentence beginning on Pg. 9, line 2 ("Both EPA (Figure 2) and SEARCH (Figure 3) data suggest that O3 mixing ratios increased during the 1990s, then began declining.") suggests that the trends in Figure 3 should be calculated only after the increase had ended, i.e., beginning in the year ~2000. When this is done, some of the trends (i.e., CTR, YRK and OAK) will be steeper, and there may be better agreement among the trends at the different sites.

It is better for us to present the full record and then add results that are restricted to the later years for comparison, since we cannot justify starting the trends at an arbitrary date. We added text to Figure 3 to describe the post-1999 trends.

6) The correlations shown in Figure 5 are misleading, and this figure should not be included without extensive modification. One major problem is that the figure combines wintertime data, when O3 concentrations may be reduced below those in transported background air due to titration by NO emissions, with summertime data, when O3 concentrations are increased above those in transported background air due to photochemical O3 production. The figure should either include data from one season only, or plot Ox (= O3 + NO2) concentrations, which are much less sensitive to the NO titration, instead of O3 only concentrations. The SEARCH data are somewhat unique in having simultaneous high quality O3 and NO2 data, and this analysis should take advantage of this uniqueness. This plot may be further confused by wintertime conversion of NOx to NOz through NO3 and N2O5 chemistry, which destroys rather than produces O3.

Figure 5 has been replaced.

7) Section 4.4 attempts to quantify ozone production efficiency (OPE) from observations, but this entire discussion must be rethought. There may be something of value in the extensive analysis that the authors performed, but the current discussion is simply not correct.

5     We rewrote Section 4.4. Section 4.4.1 is now restricted to presenting the basic regression results and demonstrating that these results exhibit temporal changes. Both the regression results and their temporal change are supported by the data. Section 4.4.2 compares regression results to those reported elsewhere, discusses the relevance of the comparisons, and examines issues raised by the reviewer.

10    Specific difficulties include:

• Ozone is quite low (≤20 ppb) at low NOz concentrations in figures 8 and 9; this immediately identifies a clear problem in the analysis. The observationally based determination of OPE implicitly assumes that "background air" contains zero NOz concentrations and O3 concentrations representing regional background transported into the region. Variations of O3 concentrations transported into the region must

15  be negligible compared to the O3 produced within the region or locally. That is simply not the case here. With few exceptions, all of the O3 concentrations in Figure 9 are <65 ppb. Berlin et al. [2013] show that regional background O3 concentrations varied between ~10 and 70 ppb in the Houston area in the mid 2000s. Thus, it is conceivable that Figure 8 and 9 (particularly the latter) are dominated by O3-NOz relationships in the transported regional background, and provide little or no information regarding ozone

20  formation within the SEARCH region.

The new discussion of transported $O_3$ in Section 4.1 examines the consistency of the intercepts shown in our figures to the literature on transported $O_3$ mixing ratios, and we attempt to identify which research efforts are the more appropriate comparisons. We revised Figure 9 to differentiate types of weather and we added discussion of the sensitivity of the regression results to variations in $O_3$ transport.

• Figure 9 gives linear fits of observed O3 vs NOz for one year, and Table S4 gives the results for all years of data. The figure below shows the relationship between the derived slopes and intercepts for all years and all sites in Table S4. If the slopes were indeed providing information about the local and regional

photochemistry, they would be expected to be independent of the intercepts, which reflect the regional background; such independence is clearly not seen. For the two urban sites (BHM and JST) the intercepts account for almost 80 of the variability in the slope.

As we had noted near the end of Section 4.4.1, intercepts and slopes are expected to be related if determined for successive tangents to a non-linear relationship. Nonlinearity is indicated in Figure 8 and in new Figure S14. The downward trends in $NO_z$ and $HNO_3$ mixing ratios indicates that older data represent tangents determined for higher mean $NO_z$ and $HNO_3$ mixing ratios.

• The paragraph beginning on pg. 12, line 7 attempts to account for the influence of depositional loss of NOz on derived OPE values, and the influence of varying background O3 concentrations. Unfortunately, the three different methods employed, yield quite different OPE values (Figures S15 – 17). Also, the results do not make good physical sense; e.g. how can OPE be near zero in 2001 at JST? Thus, this discussion increases the skepticism with which the entire analysis must be considered.

We explained why the difference-based regressions are expected to yield different results and why the statistical uncertainties are high. Please see the caveat about the 2001 data that we expressed in Section 3.1 and in the captions of Figure 8 and Figure S3. Although we could remove selected years from the analysis, we prefer to retain them with the caveats.

• The paragraph beginning on pg. 13, line 4 compares the intercepts of year-specific regressions for 2013 (~20 ppb O3) with other estimates of background levels. However, this comparison is not valid. Some of the references cited (Lefohn et al., 2014; Dolwick et al., 2015) are modeling studies that discuss U.S. background O3 according to the EPA definition, which is the O3 concentration that would exist if all U.S. anthropogenic emissions of ozone precursors were reduced to zero. Others (Chan and Vet, 2010) report observationally-based estimated baseline O3 concentrations in the absence of any continental influences. These two concepts are very different from regional background O3, i.e. the O3 concentration actually transported into the region of interest, including from other U.S. regions that are rich in anthropogenic emissions of ozone precursors. A comparison with the work of Berlin et al. [2013] is much more appropriate for discussion of the SEARCH region.

We agree that the definitions of background differ among studies. We clarified these differences and added the citation to Berlin et al. (2013).

• In Section 4.4.2 the authors compare their results with cited work from the published literature. Many of the references cited give results from studies that suffer from the same problems as plague the present work. For example Travis et al. (2016) follow much the same approach as the present paper - they interpret the slope of the correlations of Ox vs. NOz as OPE with no analysis to ensure that the low Ox-low NOz air and the high Ox-high Oz air actually represent similar background Ox and NOz concentrations, to which varying amounts of precursors were injected and subsequently photochemically processed. Reliable analysis of OPEs requires careful plume analysis, similar to that presented in Neuman et al., 2009 (a reference that is not cited in the present paper). One approach to deriving OPEs from surface site data is given by McDuffie et al., 2009 (a reference that the authors cite, but do not discuss the OPE results therein.) The references to Liu et al. (1987) and Lin et al. (1988) are not germane to the present discussion, as these results are from a very early global model, and report the total ozone produced when all VOCs, including only relatively unreactive VOCs are completely oxidized over months.

We revised this discussion.

• Finally, a very simple argument makes it quite clear that something is amiss in the entire OPE analysis. Section 4 begins with a discussion of trends in NOx emissions, emphasizing a reduction of a factor of ~3 between 1996 and 2014. Figure 10 suggests that OPE has increased by a factor of ~5. If both of these findings were correct, then O3 concentrations, at least from local and regional production, would have increased, not decreased, over this period. Yet the authors note that O3 concentrations have in fact decreased. There is a critical inconsistency buried in this analysis

We agree that this comparison raises interesting questions and have added sentences to the beginning of Section 4.4.2 that pose the argument. Figure 10 suggests slope increases of a factor of ~3 to 4 considering the actual regression slopes rather than the trendline. The difference-based regressions suggest slope increases of a factor of ~2, which are not inconsistent with the emission decreases and observed ozone

declines. However, it appears more defensible to us to present the regression results and to identify possible causes, rather than to attribute the slope changes to a single causal factor such as OPE.

• Section 4.4.3 is highly speculative, and based upon inaccurate OPEs as discussed above. It should be eliminated in its entirety, or at least extensively modified if the issues listed above can be effectively addressed.

Revised and retitled as "Implications."

7) The Conclusions section must be revised consistent with the revisions needed to address the above issues.

Revised.

Minor issues:

1) Line 11: typo - "... in in Alabama and Georgia."

Corrected.

2) In my opinion Figure 1a would be more informative as a semi-log plot. Then the NOx emission and nitrate deposition traces would parallel each other, and the linear slope of the log-transformed data would be directly proportional to the % decrease/yr. If the NOx emissions were plotted on the right axis and the deposition data on the left with the same factor change on each axis, but the offset on each axis chosen properly, then the emissions and deposition curves would be approximately superimposed.

Change made and text revised.

3) Pg. 8, lines 18-19: At least the SEARCH downward trends in mean annual HNO3 concentrations in %/yr that can be derived from Figure S4 should be compared to the corresponding trends in NOx emission and nitrate deposition. (The EPA HNO3 trends do not seem to make good physical sense.) Figure S4 also would be more informative as a semi-log plot.

We revised Figure S4 and added a statement about $HNO_3$ and $NO_y$ trends to the text.

4) The de Gouw et al., 2014 reference is omitted from the References list.

Corrected – it was there, it had just run into the previous reference due to a missing line break.

5) I do not understand the sentence beginning on Pg. 8, line 23: "Spatial variability of the annual 4th-highest daily peak 8-hour O3 mixing ratios has decreased (Figure 2), consistent with an analysis of data from a larger number of U.S. and European locations (Paoletti, et al., 2014)." Figure 2 has no direct information regarding spatial variability. It is true that the spread in the percentiles of the 4th highest O3 concentrations has decreased, but his is only to be expected as the absolute magnitude of the anthropogenic ozone enhancement has decreased. In terms of absolute ozone concentration, then the spatial variability is expected to have decreased simply because all of the region is approaching the U.S. background O3 concentration, which is expected to have small spatial variability in the Southeastern U.S. This sentence should be more clearly explained.

Revised.

6) I suggest that the sentence beginning on Pg. 9, line 7 be reworded: "The meteorological factors having the strongest influence on daily peak 8-hour O3 mixing ratios at SEARCH sites are daily maximum temperature and mid-day relative humidity (RH), whose variations cause daily peak 8-hour O3 mixing ratios to vary by ~ ±30 percent from mean peak 8-hour O3 mixing ratios (Blanchard et al., 2014)." I assume that these results are simply correlations, without proof of cause; thus the sentence should read something like: "The meteorological factors correlating most strongly with daily peak 8-hour O3 mixing ratios at SEARCH sites are daily maximum temperature and mid-day relative humidity (RH), with variations of daily peak 8-hour O3 of ~ ±30 percent from mean peak 8-hour O3 mixing ratios (Blanchard et al., 2014)."

Revised with different wording than suggested, because the statistical model was not based on linear correlations and it controlled for multiple meteorological factors.

7) The sentence beginning on Pg. 9, line 24 is likely misleading: "Background O3 may also represent an increasing absolute contribution in our study area, as multiple studies have demonstrated increasing trends in global background O3 mixing ratios." The cited studies have all focused on northern mid-latitudes, where the background O3 mixing ratios have indeed increased. However, Parrish et al. [2017b] show that increase generally ended in the early to mid 2000s. Further, Berlin et al. [2013] show that baseline ozone concentrations in air flowing into Texas from the Gulf of Mexico have not changed significantly over the 1990-2010 period. It is likely that the Gulf of Mexico inflow better represents the background ozone affecting the Southeastern U.S., which is the subject of this paper.

We added these citations as well as other references and placed an expanded discussion of background trends in the first section of the results.

8) The sentence on Pg. 10, lines 13-16 clearly refers to data over the full year. It would be more informative to include the % of the VOC reactivity due to isoprene just for the summer months when both the high isoprene and high ozone concentrations occur. Similarly, the alkene and aromatic contributions to average VOC OH reactivity for the high ozone summer season should be contrasted with the annual average numbers that are given.

Because the discussion of reactivity references previous work, and because the other referee proposed using the MIR and MOIR reactivity scales in place of $k_{OH}$ reactivity (which was what was previously published), we removed the statements about reactivity. The $k_{OH}$ reactivity results could be reproduced here, but additional computations of MIR and MOIR reactivity are beyond the scope and focus of the present manuscript. The seasonal variations of isoprene mixing ratios are evident in Figure 6 and show its importance during summer.

[revised manuscript text omitted]
₂ means, respectively. Mean NO₂ measurement uncertainty is estimated as 0.2 ppbv (1 sigma).**

---

## Referee Report (RR1)

Second review of " Ozone Response to Emission Reductions in the Southeastern United States" by Charles L. Blanchard and George M. Hidy

MS Number: acp-2017-534

**Summary:**

The paper is much improved from the first submission; all of my previous concerns have been addressed very well. The paper now presents very useful analysis of a data set that can serve as an important benchmark for future analyses of rural ozone in the southeast U.S., and throughout the country. The SEARCH data cover more than 2 decades of measurements made at eight sites in the southeastern U.S., and represent a unique record of high quality ozone and oxidized nitrogen species. I suggest that this paper be published as is (except for 3 very minor edits as noted below).

**Minor issues:**

1) Pg. 20, line 13 – Line break required to separate two run on references.

2) Figure S4 is confusing.  The caption should mention that these are semi-log plots, and it would be helpful to label the ordinate as, for example, ln (NOx Emissions in tonnes).

3) Pg. 15, line 7 – Misspelling of "larger".

---

## Author Response (AR2)

**Summary of revisions**

For convenience, the text of the reviewer is reproduced here in red, interspersed with our responses.

5  First, if not all of the NOz is measured, that will lead to a high bias.

We expanded the discussion of measurement accuracy to include the following. The SEARCH measurements of $NO_y$ were designed to capture particulate nitrate and organic nitrates, as well as NO, $NO_2$, $HNO_3$, and other oxidized nitrogen species. The $NO_y$ sampler derives from the ESE instrument discussed in Williams et al. (1998), which was one of five instruments for which measurements of $NO_y$

10  reproduced the sum of separately measured $NO_y$ species. Additional testing in 2013 showed that SEARCH $NO_y$ measurements agreed with the sum of measured mixing ratios of NO, $NO_2$, $HNO_3$, particulate nitrate, alkyl nitrates, and peroxy–alkyl nitrates (Hidy et al., 2014). The $NO_y$ measurements have therefore been shown to capture oxidized nitrogen species near both the beginning and the end of the study years. As described in the manuscript, the method for measuring $NO_2$ is $NO_2$-specific, but over

15  time the instruments utilized three different types of lamps for the photolytic conversion of $NO_2$ to NO. We therefore tested for biases in the $O_3$-$NO_z$ relationships by determining $O_3$-$HNO_3$ relationships, and concluded that similar temporal changes occurred in both sets of relationships.

Second, they present their method for trying to make sure that the background ozone is not biasing the

20  calculation. While I appreciate the effort, they really don't show that it works. (They do an analysis, but in the end, it is not very satisfying and needs a bit more analysis and justification.)

Since that supplemental analysis is limited by the available data, we expanded the discussion to cover supporting information from other studies to include the following. $O_3$ decreases driven by reductions of

25  $NO_x$ emissions between 1980 and 2014 were most pronounced in the southeastern US, where the seasonal onset of biogenic isoprene emissions and $NO_x$-sensitive $O_3$ production occurs earlier than in the northeastern U.S. (Lin et al., 2017). Lin et al. (2017) show that rising $NO_x$ emissions in Asia have increased modeled North American background $O_3$ levels (based on model simulations with zero North

American emissions) by ~0.2 ppbv $yr^{-1}$ in the southeastern U.S. in summer. The model-predicted increase in background $O_3$ in the southeastern U.S. is too small to be a systematic cause of our observed twofold (or more) increase in the slopes of summer $O_3$ versus $NO_z$. Moreover, the actual $O_3$ levels occurring in the southeastern U.S. during our study period would have been influenced by transport of air masses affected by non-zero North American emissions occurring upwind of our study area, i.e., by regional background, whose changes likely differ from changes in North American background.

Observed trends in the $5^{th}$ percentile $O_3$, have previously been used as indicators of changes in regional or continental background $O_3$ (e.g., Wilson et al., 2012). The $5^{th}$ percentile peak daily 8-hour $O_3$ mixing ratios decreased during summer at rural sites throughout the southeastern U.S. between 1988 and 2014 (Lin et al., 2017). By this measure, background $O_3$ levels were not increasing in the southeastern U.S. during our study period and therefore could not have introduced a positive bias in observed slopes of summer $O_3$ versus $NO_z$.

The trend in the $95^{th}$ percentile summer peak daily 8-hour $O_3$ mixing ratios in the southeastern US reported by Lin et al. is ~ -0.8 to -1.8 ppbv $yr^{-1}$, with downward trends occurring in other seasons as well. Our findings are comparable: between 1999 and 2014, the highest peak daily 8-hour $O_3$ mixing ratios occurring each month) declined at all SEARCH sites at statistically significant ($p < 0.01$) rates averaging ~1 ppbv $yr^{-1}$ (our Figure 3).

Anthropogenic emissions and long-range transport (long-range tropospheric + stratospheric) $O_3$ each accounted for about 40% (15 – 20 ppbv) of model-predicted $O_3$ below 1 km altitude at Huntsville, AL, during June 2013, while long-range transport accounted for ~80% of model-predicted $O_3$ above 4 km altitude (Johnson et al., 2016). Using ozonesondes that are launched on a typically weekly schedule, vertical $O_3$ mixing ratio profiles have been determined by the University of Alabama in Huntsville, Alabama, since 1999 (Newchurch et al., 2003; Johnson et al., 2016; University of Alabama, 2017; NOAA, 2017). We obtained the ozonesonde data (University of Alabama, 2017; NOAA, 2017) and identified the following statistically significant trends in the lower layers that are relatively more influenced by local and regional emissions according to Johnson et al. (2016): $-0.25 \pm 0.11$ ppbv $y^{-1}$ ($p < 0.05$) at 0.5 km, $-0.40 \pm 0.10$ ppbv $y^{-1}$ ($p < 0.0001$) at 1 km, $-0.42 \pm 0.09$ ppbv $y^{-1}$ ($p < 0.0001$) at 2 km, and $-0.57 \pm 0.13$ ppbv $y^{-1}$ in monthly averages of $O_3$ measurements made throughout the interval 1 – 2 km ($p < 0.001$). At

higher altitudes where Johnson et al. (2016) predicted that long-range transport is the dominant source of $O_3$, no trends occurred: $0.06 \pm 0.08$ ppbv $y^{-1}$ ($p > 0.1$) at 4 km and $0.09 \pm 0.19$ ppbv $y^{-1}$ ($p > 0.1$) at 8 km. The Huntsville ozonesonde data support our conclusion that changes in observed $O_3$-$NO_z$ relationships are not biased by trends in transport of background $O_3$.

The comparison of their OPE's to modeled values is of interest, but again, unsatisfying. Do the Liu et al., values of up to 80 make sense? Do OPEs of 20 for a NOx of 1 ppb make sense given their results? It would be good if they provide some critical analysis. If the OPEs increased from their values of, currently, about 20, to 80, while the NOz decreases from 1 to 0.1 ppb. Wouldn't this lead to ozone levels below

10   background and well below their asymptotic values? Please comment. When they say that for a limit of OPE approaching zero: : : Why does one presuppose such a limit? That is in contrast to Liu et al.

We revised this discussion. We added discussion of Reynolds et al. (2004), which is the modeling analysis that is most relevant to our study. We also clarified the applicability (or non-applicability) of related

15   studies. We eliminated the references to OPE that are inapplicable.

It is not apparent they are capturing all of the oxidized N in their work. How much of the organic N is measured (e.g., the fraction with the PM)? When they are using NOz, are they missing much (how much)? We expanded the discussion of measurement accuracy, as noted above.

Discussion of VOC reactivity: OH reactivity is a poorly used measure of ozone formation from VOCs. USE MIR or MOIR ([Carter, 1994])

We eliminated panel (d) of Figure 6, and simply referenced the literature on isoprene reactivity and the significance of seasonal isoprene emissions in the Southeast. The importance of isoprene emissions for

25   ozone production in the southeastern U.S. is well established (e.g., Chameides et al., 1988; Chameides and Cowling, 1995; Frost et al., 1998; Starn et al., 1998; Wiedinmyer et al., 2006; Zhang et al., 2014; Lin et al., 2017) and requires no further analysis.

Figure 2: What is the -10th %ile? Do you mean 10th %ile? (No minus)

Revised. The correct statement in the caption of Figure 2 should be "a) trends (ranges denote 90[th] and 10[th] percentile site's values)."

5  The Abstract is currently not very informative. More hard results should be provided. To say "O3 declines are less than proportional to the decreases in NOx" is obvious to most folks: : : there is a very non-zero ozone background, so you expect less than proportional. While they say OPE has increased, they don't say by how much. They don't say what are the ozone reductions. Provide some details. If I just read the abstract I would not have learned much, and would not really be included to read the article.

We revised the abstract (as well as the conclusions and a new section on implications) to provide details and explain the larger significance of the study.

The atmospheric chemistry primer (section 2.1) is too basic for the readers of ACPD. Some parts are fine
15  but assume the readers know reactions R1-R7.

We deleted the equations. We agree that this material appears extensively in the literature and does not need to be expanded in our manuscript, which is not intended as a comprehensive review.

In summary, the paper is informative, though I believe a number of modifications and further analysis are
20  required for acceptance.

We have provided a major revision.

Figure 5 has been replaced.

7) Section 4.4 attempts to quantify ozone production efficiency (OPE) from observations, but this entire discussion must be rethought. There may be something of value in the extensive analysis that the authors performed, but the current discussion is simply not correct.

We rewrote Section 4.4. Section 4.4.1 is now restricted to presenting the basic regression results and demonstrating that these results exhibit temporal changes. Section 4.4.2 compares results to those reported elsewhere, discusses the relevance of the comparisons, and examines issues raised by the reviewer.

Specific difficulties include:

• Ozone is quite low (≤20 ppb) at low NOz concentrations in figures 8 and 9; this immediately identifies a clear problem in the analysis. The observationally based determination of OPE implicitly assumes that

5 "background air" contains zero NOz concentrations and $O_3$ concentrations representing regional background transported into the region. Variations of $O_3$ concentrations transported into the region must be negligible compared to the $O_3$ produced within the region or locally. That is simply not the case here. With few exceptions, all of the $O_3$ concentrations in Figure 9 are <65 ppb. Berlin et al. [2013] show that regional background $O_3$ concentrations varied between ~10 and 70 ppb in the Houston area in the mid

10 2000s. Thus, it is conceivable that Figure 8 and 9 (particularly the latter) are dominated by $O_3$-NOz relationships in the transported regional background, and provide little or no information regarding ozone formation within the SEARCH region.

The new discussion of transported $O_3$ in Section 4.1 indicates that transported $O_3$ mixing ratios are consistent with the intercepts shown in our figures. We revised Figure 9 to differentiate types of weather.

15 We added discussion of the sensitivity of the results to variations in $O_3$ transport.

• Figure 9 gives linear fits of observed $O_3$ vs NOz for one year, and Table S4 gives the results for all years of data. The figure below shows the relationship between the derived slopes and intercepts for all years and all sites in Table S4. If the slopes were indeed providing information about the local and regional

20 photochemistry, they would be expected to be independent of the intercepts, which reflect the regional background; such independence is clearly not seen. For the two urban sites (BHM and JST) the intercepts account for almost 80 of the variability in the slope.

As we had noted near the end of Section 4.4.1, intercepts and slopes are expected to be related if determined for successive tangents to a non-linear relationship. Nonlinearity is indicated in Figure 8 and

25 in new Figure S14. The downward trends in $NO_z$ and $HNO_3$ mixing ratios means that older data represent tangents determined for higher mean $NO_z$ and $HNO_3$ mixing ratios.

• The paragraph beginning on pg. 12, line 7 attempts to account for the influence of depositional loss of NOz on derived OPE values, and the influence of varying background O3 concentrations. Unfortunately, the three different methods employed, yield quite different OPE values (Figures S15 – 17). Also, the results do not make good physical sense; e.g. how can OPE be near zero in 2001 at JST? Thus, this discussion increases the skepticism with which the entire analysis must be considered.

We explained why the difference-based regressions are expected to yield different results and why the statistical uncertainties are high. Please see the caveat about the 2001 data that we expressed in Section 3.1 and in the captions of Figure 8 and Figure S3. Although we could remove selected years from the analysis, we prefer to retain them with the caveats.

• The paragraph beginning on pg. 13, line 4 compares the intercepts of year-specific regressions for 2013 (~20 ppb O3) with other estimates of background levels. However, this comparison is not valid. Some of the references cited (Lefohn et al., 2014; Dolwick et al., 2015) are modeling studies that discuss U.S. background O3 according to the EPA definition, which is the O3 concentration that would exist if all U.S. anthropogenic emissions of ozone precursors were reduced to zero. Others (Chan and Vet, 2010) report observationally-based estimated baseline O3 concentrations in the absence of any continental influences. These two concepts are very different from regional background O3, i.e. the O3 concentration actually transported into the region of interest, including from other U.S. regions that are rich in anthropogenic emissions of ozone precursors. A comparison with the work of Berlin et al. [2013] is much more appropriate for discussion of the SEARCH region.

We agree that the definitions of background differ among studies. We clarified these differences and added the citation to Berlin et al. (2013).

• In Section 4.4.2 the authors compare their results with cited work from the published literature. Many of the references cited give results from studies that suffer from the same problems as plague the present work. For example Travis et al. (2016) follow much the same approach as the present paper - they interpret the slope of the correlations of Ox vs. NOz as OPE with no analysis to ensure that the low Ox-low NOz air and the high Ox-high Oz air actually represent similar background Ox and NOz concentrations, to

which varying amounts of precursors were injected and subsequently photochemically processed. Reliable analysis of OPEs requires careful plume analysis, similar to that presented in Neuman et al., 2009 (a reference that is not cited in the present paper). One approach to deriving OPEs from surface site data is given by McDuffie et al., 2009 (a reference that the authors cite, but do not discuss the OPE results therein.) The references to Liu et al. (1987) and Lin et al. (1988) are not germane to the present discussion, as these results are from a very early global model, and report the total ozone produced when all VOCs, including only relatively unreactive VOCs are completely oxidized over months.

We revised this discussion. We expanded the discussion of modeling work by Reynolds et al. (2004), which is directly relevant to our study region and time period.

• Finally, a very simple argument makes it quite clear that something is amiss in the entire OPE analysis. Section 4 begins with a discussion of trends in NOx emissions, emphasizing a reduction of a factor of ~3 between 1996 and 2014. Figure 10 suggests that OPE has increased by a factor of ~5. If both of these findings were correct, then O3 concentrations, at least from local and regional production, would have increased, not decreased, over this period. Yet the authors note that O3 concentrations have in fact decreased. There is a critical inconsistency buried in this analysis

We added comments on this matter to the beginning of Section 4.4.2.

• Section 4.4.3 is highly speculative, and based upon inaccurate OPEs as discussed above. It should be eliminated in its entirety, or at least extensively modified if the issues listed above can be effectively addressed.

Revised and retitled.

7) The Conclusions section must be revised consistent with the revisions needed to address the above issues.

Revised.

Minor issues:

1) Line 11: typo - "... in in Alabama and Georgia."

Corrected.

2) In my opinion Figure 1a would be more informative as a semi-log plot. Then the NOx emission and nitrate deposition traces would parallel each other, and the linear slope of the log-transformed data would be directly proportional to the % decrease/yr. If the NOx emissions were plotted on the right axis and the deposition data on the left with the same factor change on each axis, but the offset on each axis chosen properly, then the emissions and deposition curves would be approximately superimposed.

Change made and text revised.

3) Pg. 8, lines 18-19: At least the SEARCH downward trends in mean annual HNO3 concentrations in %/yr that can be derived from Figure S4 should be compared to the corresponding trends in NOx emission and nitrate deposition. (The EPA HNO3 trends do not seem to make good physical sense.) Figure S4 also would be more informative as a semi-log plot.

We revised Figure S4 and added a statement about $HNO_3$ and $NO_y$ trends to the text.

4) The de Gouw et al., 2014 reference is omitted from the References list.

Corrected – it was there, it had just run into the previous reference due to a missing line break.

5) I do not understand the sentence beginning on Pg. 8, line 23: "Spatial variability of the annual 4th-highest daily peak 8-hour O3 mixing ratios has decreased (Figure 2), consistent with an analysis of data from a larger number of U.S. and European locations (Paoletti, et al., 2014)." Figure 2 has no direct information regarding spatial variability. It is true that the spread in the percentiles of the 4th highest O3 concentrations has decreased, but his is only to be expected as the absolute magnitude of the anthropogenic ozone enhancement has decreased. In terms of absolute ozone concentration, then the spatial variability is expected to have decreased simply because all of the region is approaching the U.S. background O3 concentration, which is expected to have small spatial variability in the Southeastern U.S. This sentence should be more clearly explained.

Revised.

6) I suggest that the sentence beginning on Pg. 9, line 7 be reworded: "The meteorological factors having the strongest influence on daily peak 8-hour O3 mixing ratios at SEARCH sites are daily maximum temperature and mid-day relative humidity (RH), whose variations cause daily peak 8-hour O3 mixing ratios to vary by ~ ±30 percent from mean peak 8-hour O3 mixing ratios (Blanchard et al., 2014)." I assume that these results are simply correlations, without proof of cause; thus the sentence should read something like: "The meteorological factors correlating most strongly with daily peak 8-hour O3 mixing ratios at SEARCH sites are daily maximum temperature and mid-day relative humidity (RH), with variations of daily peak 8-hour O3 of ~ ±30 percent from mean peak 8-hour O3 mixing ratios (Blanchard et al., 2014)."

Revised with different wording than suggested, because the statistical model was not based on linear correlations and it controlled for multiple meteorological factors.

7) The sentence beginning on Pg. 9, line 24 is likely misleading: "Background O3 may also represent an increasing absolute contribution in our study area, as multiple studies have demonstrated increasing trends in global background O3 mixing ratios." The cited studies have all focused on northern mid-latitudes, where the background O3 mixing ratios have indeed increased. However, Parrish et al. [2017b] show that increase generally ended in the early to mid 2000s. Further, Berlin et al. [2013] show that baseline ozone concentrations in air flowing into Texas from the Gulf of Mexico have not changed significantly over the 1990-2010 period. It is likely that the Gulf of Mexico inflow better represents the background ozone affecting the Southeastern U.S., which is the subject of this paper.

We added these citations as well as other references and placed an expanded discussion of background trends in the first section of the results.

8) The sentence on Pg. 10, lines 13-16 clearly refers to data over the full year. It would be more informative to include the % of the VOC reactivity due to isoprene just for the summer months when both the high isoprene and high ozone concentrations occur. Similarly, the alkene and aromatic contributions

to average VOC OH reactivity for the high ozone summer season should be contrasted with the annual average numbers that are given.

Because the discussion of reactivity references previous work, and because the other referee proposed using the MIR and MOIR reactivity scales in place of $k_{OH}$ reactivity (which was what was previously published), we removed the statements about reactivity. The $k_{OH}$ reactivity results could be reproduced here, but additional computations of MIR and MOIR reactivity are beyond the scope and focus of the present manuscript. The seasonal variations of isoprene mixing ratios are evident in Figure 6 and show its importance during summer.

[revised manuscript text omitted]

---

## Author Response (AR3)

All requested technical corrections have been made. They are marked in red. We also corrected the caption and y-axis labels in Figure S4 as suggested.

[revised manuscript text omitted]